# MENLO🌍: FROM PREFERENCES TO PROFICIENCY – EVALUATING AND MODELING NATIVE-LIKE QUALITY ACROSS 47 LANGUAGES

**Chenxi Whitehouse**[†]    **Sebastian Ruder**[†]    **Tony Zhiyang Lin**    **Oksana Kurylo**
**Haruka Takagi**    **Janice Lam**    **Nicolò Busetto**    **Denise Diaz**

Meta Superintelligence Labs
chenxwh@meta.com    ruder@meta.com

## ABSTRACT

Ensuring native-like quality of large language model (LLM) responses across many languages is challenging. To address this, we introduce MENLO, a framework that operationalizes the evaluation of native-like response quality based on audience design-inspired mechanisms. Using MENLO, we create a dataset of 6,423 human-annotated prompt–response preference pairs covering four quality dimensions with high inter-annotator agreement in 47 language varieties. Our evaluation reveals that zero-shot LLM judges benefit significantly from pairwise evaluation and our structured annotation rubrics, yet they still underperform human annotators on our dataset. We demonstrate substantial improvements through fine-tuning with reinforcement learning, reward shaping, and multi-task learning approaches. Additionally, we show that RL-trained judges can serve as generative reward models to enhance LLMs' multilingual proficiency, though discrepancies with human judgment remain. Our findings suggest promising directions for scalable multilingual evaluation and preference alignment. We release our dataset and evaluation framework to support further research in multilingual LLM evaluation.

🤗 **Dataset** https://huggingface.co/datasets/facebook/menlo

## 1 INTRODUCTION

In order for LLMs to be most useful across the globe, they need to be able to provide high-quality responses in many languages. Responses should be relevant (Zhuang et al., 2024), factually accurate (Jacovi et al., 2025), and natural (Marchisio et al., 2024; Guo et al., 2025), among other considerations. Ultimately, for interaction in any language to be seamless, responses need to be indistinguishable from those of a native speaker (Novikova et al., 2016; Liu et al., 2021). Language proficiency in humans has traditionally been evaluated via standardized tests (Jamieson et al., 2000). While such tests have been applied to evaluating LLMs (Anil et al., 2023; Mayor-Rocher et al., 2024; Lothritz & Cabot, 2025), they are difficult to scale and do not readily correspond to real-world conversations. What is considered a *native-like* response largely depends on speakers' and listeners' interpretations of whom they are speaking to (Bell, 1984).

To operationalize the evaluation of native-like response quality across languages, we propose **M**ultilingual **E**valuation of **N**ative-**L**ike **O**utput (MENLO); see Figure 1 for an overview. MENLO breaks down native-like response quality into four key dimensions: i) language quality and coherence; ii) alignment with cultural and linguistic nuances of a specific language variety/locale; iii) factual correctness and grounding in the local context; and iv) overall writing style and helpfulness.

Building on mechanisms from audience design (Bell, 1984), we propose creating tailored prompts that effectively evoke local contexts by defining the target audience (e.g., an addressee or reference group), thereby guiding the generated language to converge to contextually appropriate "native"

---

[†]Equal Contribution.

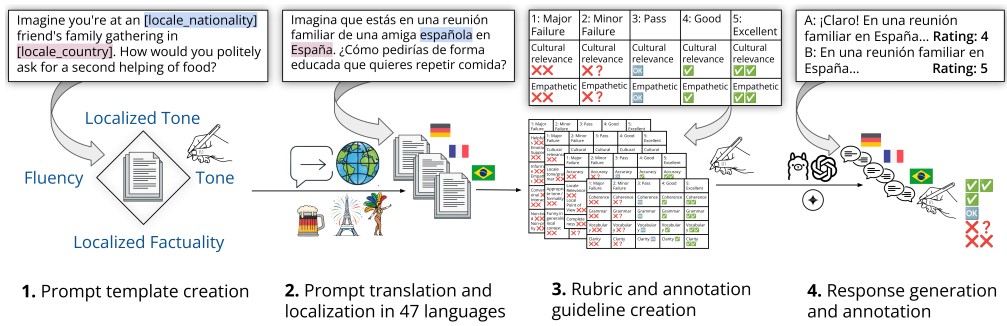

Figure 1: MENLO framework and annotation process. 1) Human-written prompt templates evoking local contexts are created in English for the four dimensions. 2) Prompt templates are translated and localized into 47 language varieties. 3) Annotation guidelines are created that break down each dimension into easy-to-follow rubrics. 4) LLMs are used to generate response pairs for each prompt, which are annotated with Likert-scale ratings and preferences.

styles. We develop instructions that reduce annotation subjectivity and improve inter-annotator agreement. Responses are generated using state-of-the-art LLMs and annotated with ratings on a 1–5 Likert scale, with an average Krippendorff's $\alpha = 0.84$. Overall, the MENLO dataset consists of 6,423 annotated prompt-response preference pairs, and 81,014 annotations, in 47 language varieties.

Human evaluation, particularly at a massively multilingual scale is expensive. We thus evaluate the ability of LLMs to serve as judges of native-like quality responses. We find that in zero-shot setting, **pairwise evaluation**—where models predict scores for two responses simultaneously (without explicitly predicting preference)—significantly outperforms its pointwise counterpart. The advantage of evaluating two responses side-by-side is even bigger than in-context examples with labels. In addition, we observe significant improvements with judges using our annotation rubrics compared to judges without rubrics, highlighting the generality of the MENLO framework.

As zero-shot judges remain below human annotation quality, using the pairwise evaluation setup, we fine-tune `Qwen3-4B` and `Llama4-Scout` as LLM judges on the MENLO training data, finding that RL-trained models outperform their SFT counterparts. In particular, a multi-task `Llama4-Scout` model trained with shaped rewards surpasses frontier API models with the strongest overall performance across 47 language varieties, reaching agreement levels comparable to human annotators.

Finally, we demonstrate that these judges can be used as generative reward models (RMs) to directly improve a policy model's proficiency. By using our pairwise RL-trained `Qwen3-4B` judge to post-train the base `Qwen3-4B` model, we observe quality gains as measured by both LLM evaluators and human raters. However, LLM evaluators tend to be overconfident in assessing improvements compared to human judgments (+0.6 higher gain). This finding shows that while judges trained with our framework can successfully drive model improvements, the gap between LLM and human raters highlights the remaining challenges in reliably modeling native-like quality across languages.

Our contributions are the following: **1)** We develop MENLO, a framework for the evaluation of native-like response quality in four dimensions, based on principles from audience design, employing parametric templates and carefully crafted annotation guidelines. **2)** We create the MENLO dataset, consisting of 6,423 annotated prompt-response preference pairs in 47 language varieties. **3)** We evaluate zero-shot judges on the annotated data, demonstrating the benefits of pairwise evaluation and rubrics. **4)** We show that multi-task RL and reward shaping enables fine-tuning a judge that is on par with human annotators in 47 language varieties. **5)** We demonstrate that pairwise fine-tuned judges can be used as generative RMs to improve policy model language proficiency, while we find that LLM evaluations tend to overestimate improvements compared to human raters.

Our framework unifies the MENLO dataset, RL-trained pairwise judging, and generative reward modeling, offering a practical and scalable approach to both assess and improve native-like quality.

| Dimension | Definition | Key Question | Example Prompt |
|---|---|---|---|
| **Fluency** | Language proficiency compared to an expert-level native speaker. | Is the model response coherent, well-versed, clear, and free from grammatical errors? | If you could make one change to the education system in your country, what would it be and why? |
| **Tone** | Overall writing style or "voice" of the response. | Is the response helpful, insightful, engaging, and fair? | I'm so lost, I feel like I've been stuck in neutral for weeks. |
| **Localized Tone** | Alignment with cultural, regional, and linguistic nuances. | Does the response employ locally relevant expressions and is culturally sensitive? | Imagine you're at an [locale_nationality] friend's family gathering in [locale_country]. How would you politely ask for a second helping of food? |
| **Localized Factuality** | Factuality, completeness, and grounding in local context. | Is the response factual, complete, and grounded in the local context? | You are staying with a host family in [locale_country] during [locale_holiday]. They invite you to help with the preparations. What tasks might you be expected to help with, and what do they symbolize? |

Figure 2: Dimensions of native-like response quality in MENLO and example prompt (template).

Table 1: Annotation and statistics of MENLO across evaluation dimensions. IAA presents *Krippendorff's* $\alpha$ measuring inter-annotator agreement. Average token counts are computed using Qwen3-4B.

| Dimension | # Annotations | # Annotators | # Prompts | Avg # Tokens | | Rating (1–5 Scale) | | |
|---|---|---|---|---|---|---|---|---|
| | | | | Prompt | Response | IAA | Mean | Std. |
| Fluency | 23,556 | 450 | 1,820 | 81.6 | 804.3 | 0.82 | 4.01 | 1.11 |
| Tone | 18,712 | 429 | 1,410 | 27.8 | 575.3 | 0.86 | 3.48 | 1.35 |
| Localized Tone | 22,324 | 530 | 1,815 | 71.7 | 559.2 | 0.83 | 3.89 | 1.17 |
| Localized Factuality | 16,422 | 525 | 1,378 | 121.8 | 839.1 | 0.84 | 3.82 | 1.16 |
| **Overall** | 81,014 | 1,934 | 6,423 | 75.6 | 692.2 | 0.84 | 3.80 | 1.20 |

## 2 THE MENLO DATASET

MENLO characterizes native-like conversational response quality in a language along four key dimensions: fluency, tone, localized tone, and localized factuality. These dimensions go beyond prior work that focused mainly on naturalness (Novikova et al., 2016; Liu et al., 2021; Guo et al., 2025) and are motivated by work on language proficiency assessment (Ke & Ng, 2019), cross-cultural variation (Hershcovich et al., 2022; Myung et al., 2024), local knowledge grounding (Hupkes & Bogoychev, 2025). We provide further context on our definition of these dimensions in Figure 2.

From a sociolinguistic perspective, the Style Axiom (Bell, 1984) states that intraspeaker variation (style) reflects interspeaker variation (social). Native-like quality is therefore not a single fixed target but a socially conditioned range of stylistic choices that depend on interlocutors. Key mechanisms include accommodation, where speakers adapt their style to the addressee, and referee design, where speakers align with an absent reference group they wish to identify with. These mechanisms motivate our focus on tone and localized tone as central to native-like quality. To operationalize these ideas, we design human-written parametric English prompt templates for each dimension with placeholders such as [locale_nationality], [locale_country], [locale_holiday], etc. By defining the addressee or reference group, these prompts evoke local contexts and guide models toward contextually appropriate "native" styles. We provide an overview of the MENLO framework and annotation process in Figure 1.

We select 47 language varieties representing a typologically diverse set of widely used languages and their major variants, including, e.g., South American and European varieties of Spanish and Portuguese, several varieties of English, and romanized versions of non-Latin script languages (see Appendix A.2). Native speakers are recruited to professionally translate these prompt templates, with placeholders instantiated using locally relevant entities. As native quality is tied to the local context, we ensure that native speakers are from the specific regions where the corresponding language varieties are spoken. Similar criteria are used to select annotators for each language variety. Each language variety has approximately the same number of examples in MENLO.

To ensure consistency in evaluation, we develop instructions that reduce the subjectivity of the annotation and break down the four broad dimensions into easy-to-follow rubrics and self-explanatory signals (human-written). Annotators receive guidelines with examples for each dimension. We additionally develop a customized annotation tool and annotator screening tests to filter out unreliable

Table 2: Comparison of multilingual response quality datasets: RECON (Doddapaneni et al., 2025), PARIKSHA (Watts et al., 2024), BIGGEN BENCH (Kim et al., 2025), M-REWARDBENCH (Gureja et al., 2025), MM-EVAL (Son et al., 2024). $|\mathcal{L}|$: # of languages, $|\mathcal{D}|$: # of prompts, IAA: inter-annotator agreement, IF: instruction following. *: 81,014 annotations; 1,776 test examples.

| Dataset | $|\mathcal{L}|$ | $|\mathcal{D}|$ | IAA | Dimensions | Prompts | Responses | Ratings |
|---|---|---|---|---|---|---|---|
| MENLO | 47 | 6,423* | 0.84 | Fluency, tone, localized tone, localized factuality | Human-written, translated & localized | Annotated in each language | Preference & 1–5 |
| PARIKSHA | 10 | 200 | 0.54 | Hallucinations, task quality, linguistic acceptability | Human-written | Annotated in each language | Preference & 0–2 |
| BIGGEN BENCH | 10 | 420 | – | Poem, reasoning, humor, translation, historical text | Human-written, LLM-augmented | Generated in each language | 1–5 |
| RECON | 6 | 3,000 | – | IF, theory of mind, reasoning, safety, planning, etc. | Translated | Generated in each language | Preference & 1–5 |
| MM-EVAL | 18 | 4,981 | – | Reasoning, chat, linguistics, hallucination, safety | Translated | Generated in each language | Preference |
| M-REWARD BENCH | 23 | 66,787 | – | Chat, safety, reasoning, translation | Translated | Translated | Preference |

annotators. Furthermore, we train 1–2 expert annotators per language who provide language-specific feedback to annotators and provide gold annotations on a subset of examples.

We generate two responses for each prompt with state-of-the-art LLMs including GPT-4o, Llama4-Maverick, Llama4-Maverick with Search, and Gemini 1.5 with Search. We present both responses in randomized order to human annotators and ask them to provide 1–5 Likert ratings per response, allowing ties. Each response pair is annotated by at least 3 annotators, with final scores aggregated via majority vote. Annotators achieve high reliability, with an average Krippendorff's $\alpha = 0.84$.

Overall, MENLO consists of 6,423 annotated prompt-response preference pairs across 47 language varieties, each containing a prompt, two responses, and corresponding scores, totaling 81,014 human annotations. Summary statistics are reported in Table 1. Example prompts per dimension are shown in Figure 2. Further details of MENLO including annotation process, language coverage, rubrics, and full examples featuring responses and their corresponding ratings are provided in Appendix A.

Table 2 compares MENLO with existing multilingual preference datasets. MENLO provides localized prompts and responses, spans more languages, and reaches higher agreement than prior work.

## 3 EVALUATING LLM-JUDGES ON MENLO

We next evaluate the ability of LLMs to serve as automatic judges of native-like quality on MENLO. Out of the 6,423 pairs, we hold out 1,766 pairs (3,552 responses) as the test set,[1] and use the remainder for training and prompt development (see §4 and §5). Where expert annotations are available, we use these as labels. For the remaining responses, we average the annotated ratings of each response.[2] Our evaluation focuses on three questions: (i) how pointwise and pairwise setups compare, (ii) the effect of few-shot exemplars, and (iii) the role of explicit grading rubrics.

We benchmark a range of open-source and API-based models, covering both *thinking* and *non-thinking* variants: `Qwen3-4B`, `Qwen3-32B`, `Llama-3.1-8B`,[3] `Llama-3.3-70B`, `Llama4-Scout`, `o3`, `gpt-4o`, and `gpt-4.1`. All models are used in the default setup with maximum output length 8192.

We report two primary metrics: (i) *Macro-F1* for 5-way classification, and (ii) *Preference* accuracy over Win/Loss/Tie outcomes. Note that *we do NOT directly ask for preference judgments; rather, we infer these from the assigned grades*. Additionally, we report classification accuracy, Krippendorff's $\alpha$, which measures agreement with human annotators while accounting for chance agreement and missing data, and provide detailed per-dimension and per-language breakdowns in Appendix E.

---

[1] Translations of the same prompt template are assigned the same set to prevent train-test leakage.

[2] Multiple annotations can be used in future work on pluralistic alignment (Sorensen et al., 2024).

[3] Llama models are instruction-tuned and we omit the `Instruct` suffix for brevity.

Table 3: Zero-shot and few-shot results of open-source and API models on the MENLO test set using **POINTWISE** (grading single responses) and **PAIRWISE** (grading response pairs) scoring (see §3.1). *Macro-F1* shows 5-way classification performance and *Preference* reports accuracy on Win/Loss/Tie. Reported gains/loss are relative to zero-shot pointwise performance.

| | Macro F1 | | | Preference Accuracy | | |
|---|---|---|---|---|---|---|
| **MODELS** | ZERO-SHOT **POINTWISE** | FEW-SHOT **POINTWISE** | ZERO-SHOT **PAIRWISE** | ZERO-SHOT **POINTWISE** | FEW-SHOT **POINTWISE** | ZERO-SHOT **PAIRWISE** |
| Qwen3-4B | 23.06 | 31.18 $_{+8.12}$ | 35.46 $_{+12.40}$ | 40.54 | 39.35 $_{-1.19}$ | 57.13 $_{+16.57}$ |
| Qwen3-32B | 28.53 | 35.45 $_{+6.92}$ | 37.48 $_{+8.95}$ | 42.19 | 42.87 $_{+0.68}$ | 59.12 $_{+16.59}$ |
| Llama-3.1-8B | 22.27 | 23.29 $_{+1.02}$ | 29.46 $_{+7.19}$ | 39.92 | 37.15 $_{-2.77}$ | 50.45 $_{+10.48}$ |
| Llama-3.3-70B | 27.93 | 30.52 $_{+2.59}$ | 37.50 $_{+9.57}$ | 37.37 | 38.56 $_{+1.19}$ | 55.32 $_{+17.89}$ |
| Llama4-Scout | 25.63 | 32.84 $_{+7.21}$ | 36.11 $_{+10.48}$ | 42.19 | 41.22 $_{-0.97}$ | 56.25 $_{+14.12}$ |
| o3 | 26.54 | 27.92 $_{+1.38}$ | 35.35 $_{+8.81}$ | 45.07 | 44.68 $_{-0.39}$ | 58.72 $_{+13.68}$ |
| gpt-4o | 25.99 | 29.57 $_{+3.58}$ | 37.57 $_{+11.58}$ | 42.92 | 45.87 $_{+2.95}$ | 57.98 $_{+15.09}$ |
| gpt-4.1 | 32.23 | 33.84 $_{+1.61}$ | 38.53 $_{+6.30}$ | 41.73 | 44.00 $_{+2.27}$ | 59.23 $_{+17.50}$ |

## 3.1 POINTWISE VS. PAIRWISE

Although MENLO provides paired responses for each prompt, the presence of detailed *grading rubrics* means that *pointwise* evaluation is in principle sufficient: a model could *assign absolute scores to individual responses without needing comparisons*. However, pairwise setups may provide stronger relative signals by anchoring judgments against another candidate. We therefore compare three setups: **Zero-shot pointwise**: the model is given a prompt, a *single* response, and a detailed 5-point grading rubric, and asked to generate evaluation reasoning (in *thinking*) and assign a final grade; **Few-shot pointwise**: we additionally provide three graded examples: one from 1–2, one with a grade of 3, and one from 4–5; **Zero-shot pairwise**: the model is presented with *both* responses to the same prompt and asked to assign a grade to each, following the template in Figure 17 (Appendix C), without constraints on ties. The order of the two responses is randomized.

Table 3 reports Macro-F1 and Preference results. Zero-shot pairwise consistently outperforms both zero-shot and few-shot pointwise scoring across models, with gains of up to $+12.4\%$ in Macro-F1 and $+18.0\%$ in Preference accuracy over zero-shot pointwise. Few-shot pointwise improves Macro-F1 relative to zero-shot pointwise but yields only marginal gains in Preference, still falling short of zero-shot pairwise by an *average* of $-5.5\%$ in Macro-F1 and $-15.1\%$ in Preference across models.

These results indicate that models are substantially more reliable at assigning scores when evaluating two responses side by side, even without ground-truth labels. The unexpectedly large gains over few-shot in-context examples highlight **pairwise evaluation**, which explicitly anchors outputs against a competing candidate (Wang et al., 2025), as a promising direction for improving automated judging reliability, even when the ultimate goal is pointwise scoring. We also evaluate few-shot pairwise on Qwen3-4B, observing only a small gain in Macro-F1 ($+0.6$) relative to zero-shot pairwise, further supporting our findings. Future work may investigate whether extending pairwise comparisons to a listwise evaluation of multiple responses offers additional benefits.

## 3.2 WITH AND WITHOUT GRADING RUBRICS

We further examine the role of detailed grading rubrics in judge performance. All rubrics are human-written 5-point guidelines specific to the dimension and question type of each prompt. Examples of dimension-specific rubrics are shown in Appendix A.3.

Table 4 compares zero-shot pointwise and pairwise performance with and without access to rubrics. The latter shows only the five class labels, without accompanying criteria or definitions. Results show that rubrics provide a substantial benefit, especially for pointwise evaluation, yielding average gains of $+4.3\%$ in Macro-F1 and $+2.5\%$ in Preference accuracy. In contrast, pairwise evaluation benefits more modestly, with improvements of roughly $+1\%$ on both metrics.

These findings suggest that judges perform better when grounded, either by explicit rubrics or by comparison with another response. Since pairwise comparison itself offers a strong grounding sig-

Table 4: Zero-shot performance comparing without and with detailed 5-Point Grading Rubrics.

| | Macro F1 | | | | Preference Accuracy | | | |
|---|---|---|---|---|---|---|---|---|
| **MODELS** | **POINTWISE** | | **PAIRWISE** | | **POINTWISE** | | **PAIRWISE** | |
| | *wo/ Rubrics* | *w/ Rubrics* | *wo/ Rubrics* | *w/ Rubrics* | *wo/ Rubrics* | *w/ Rubrics* | *wo/ Rubrics* | *w/ Rubrics* |
| Qwen3-4B | 16.00 | 23.06 +7.06 | 32.74 | 35.46 +2.72 | 33.52 | 40.54 +7.02 | 54.08 | 57.13 +3.05 |
| Qwen3-32B | 25.59 | 28.53 +2.94 | 38.10 | 37.48 -0.62 | 43.32 | 42.19 -1.13 | 59.23 | 59.12 -0.11 |
| Llama-3.1-8B | 21.50 | 22.27 +0.77 | 30.89 | 29.46 -1.43 | 38.34 | 39.92 +1.58 | 49.55 | 50.45 +0.90 |
| Llama-3.3-70B | 22.71 | 27.93 +5.22 | 35.12 | 37.50 +2.38 | 34.54 | 37.37 +2.83 | 56.29 | 55.32 -0.97 |
| Llama4-Scout | 22.15 | 25.63 +3.48 | 35.21 | 36.11 +0.90 | 41.28 | 42.19 +0.91 | 55.10 | 56.25 +1.15 |
| o3 | 25.43 | 26.54 +1.11 | 35.60 | 35.35 -0.25 | 45.13 | 45.07 -0.06 | 57.98 | 58.72 +0.74 |
| gpt-4o | 22.45 | 25.99 +3.54 | 36.74 | 37.57 +0.83 | 37.60 | 42.92 +5.32 | 56.85 | 57.98 +1.13 |
| gpt-4.1 | 22.26 | 32.23 +9.97 | 37.35 | 38.53 +1.18 | 38.67 | 41.73 +3.06 | 56.96 | 59.23 +2.27 |

Table 5: PAIRWISE SFT, RL, and SFT+RL-trained Qwen3-4B and Llama4-Scout results. RL-trained models perform best overall.

| PAIRWISE | Qwen3-4B | | Llama4-Scout | |
|---|---|---|---|---|
| | Marco-F1 | Preference | Marco-F1 | Preference |
| ZERO-SHOT | 35.46 | 57.13 | 36.11 | 56.25 |
| SFT | 33.44 -2.02 | 53.68 -3.45 | 44.17 +8.06 | 60.08 +3.83 |
| RL | **39.44** +3.98 | **60.02** +2.89 | 45.62 +9.51 | **62.60** +6.35 |
| SFT + RL | 39.33 +3.87 | 58.78 +1.65 | **45.82** +9.71 | 61.10 +4.85 |

Table 6: Ablation of different reward designs for PAIRWISE RL-trained Qwen3-4B. *Smooth.* and *Prefer.* refer to *Reward Smoothing* and *Preference Bonus*. See §4.1 for details.

| **REWARDS** | **Marco-F1** | **Preference** |
|---|---|---|
| *Binary Only* | 37.11 | 58.27 |
| *Binary + Smooth.* | 37.30 +0.19 | 51.47 -6.80 |
| *Binary + Prefer.* | 37.05 -0.06 | **60.48** +2.21 |
| *Binary + Smooth. + Prefer.* | **39.44** +2.33 | 60.02 +1.75 |

nal, it sees limited impact from rubrics. This highlights the importance of high-quality rubrics: if judges could automatically generate and evaluate high-quality, detailed rubrics, we hypothesize that the performance gap between pairwise and pointwise evaluation would further narrow.

## 4 TRAINING LLM-JUDGES ON MENLO

Having established in §3 that pairwise evaluation yields substantial advantages over pointwise scoring, we next examine whether training LLMs as judges can further close the gap to human annotators. We train on the MENLO training split (total 4,675 response pairs, where 232 pairs are held out for validation) and explore different learning strategies, model families, and reward designs. Inspired by the success of recent reasoning-based judges such as J1 (Whitehouse et al., 2026), we compare supervised fine-tuning (SFT) and reinforcement learning (RL), as well as single-task (dimension-specific) and multi-task (all dimensions) training.

We fine-tune two contrasting models: Qwen3-4B (dense, reasoning-oriented) and Llama4-Scout (Mixture-of-Experts, non-reasoning), which differ in architecture and cognitive approach. For **SFT**, models directly predict 5-point grades using cross-entropy loss under teacher forcing, without intermediate reasoning generation. For **RL**, we use GRPO (Shao et al., 2024) with the template from Figure 17, encouraging step-by-step reasoning before score assignment. Following Whitehouse et al. (2026), we augment training data by including both response orders (A,B) and (B,A) to mitigate positional bias. Training details are provided in Appendix D.1.

### 4.1 REWARD DESIGNS FOR RL

To make RL training effective, we design a composite reward signal that combines absolute accuracy with relative preference alignment and robustness to near-miss predictions: (i) **Pointwise binary reward**: $+1$ if the predicted score matches the gold label, 0 otherwise. (ii) **Reward smoothing**: partial reward ($+0.5$) if the prediction differs by exactly one grade. (iii) **Preference bonus**: additional $+1$ if the *sign* of the difference between the two predicted scores matches the label. (iv) **Penalties**: $-1$ for invalid or missing scores, and $-0.2$ for formatting violations, i.e. each tag must appear in the correct order and only once.

All reward components are summed to produce the final RL signal. Formally, the reward can be expressed as follows, where $s$ and $gt$ represent predicted and ground truth grades, respectively:

$$R = \sum_{i \in \{A,B\}} \underbrace{\max \left( \mathbf{1}[s_i = gt_i],\ 0.5 \cdot \mathbf{1}[|s_i - gt_i| = 1] \right)}_{\text{pointwise binary reward w/ reward smoothing}} + \underbrace{\mathbf{1}[\text{sign}(s_A - s_B) = \text{sign}(gt_A - gt_B)]}_{\text{preference bonus}}$$
$$- \underbrace{\mathbf{1}[\text{failed extraction}]}_{\text{extraction penalty}} - \underbrace{0.2 \cdot \mathbf{1}[\text{formatting violation}]}_{\text{format penalty}}.$$

## 4.2 Overall Performance: SFT vs. RL

We first compare the overall performance of SFT and RL-trained models. Table 5 shows that RL-trained `Qwen3-4B` and `Llama4-Scout` consistently outperform their SFT counterparts. For inherently thinking models like `Qwen3-4B`, SFT without Chain-of-Thought (CoT) reasoning actually hurts performance, causing a $-2.0\%$ drop in Macro-F1 and $-3.5\%$ in Preference accuracy. In contrast, RL, which incentivizes reasoning, improves performance by $+4.0\%$ in Macro-F1 and $+2.9\%$ in Preference, surpassing the best frontier API model `gpt-4.1`.

For non-thinking models like `Llama4-Scout`, SFT already provides substantial gains ($+8.1\%$ in Macro-F1 and $+3.8\%$ in Preference) compared to zero-shot. RL training further improves results, particularly in Preference ($+2.5\%$). This demonstrates the promise of pairwise RL training across model families, scales, and reasoning capabilities.

We also experimented with initializing RL from the best SFT checkpoint, but observed little or no improvement over starting RL from scratch. Models trained on SFT without CoT tend to copy the placeholder "<think> *Your analysis and reasoning here.* </think>" from the prompt rather than generating meaningful reasoning, which limits the benefit of RL. This suggests that for tasks requiring reasoning, it is preferable to start RL directly when the SFT target lacks CoT supervision.

## 4.3 Ablation of RL Rewards

In Table 6, we ablate the RL reward design to validate the contribution of each reward component in RL training: (i) *binary only*: reward $+1$ for exact score match, $0$ otherwise; (ii) *binary+smooth.*: adds partial reward for near-miss scores, no preference bonus; (iii) *binary+prefer.*: includes preference reward, no smoothing; and (iv) *binary+smooth.+prefer.*: the default reward design in §4.1. Results show clear benefits from combining reward smoothing and preference bonus, achieving the best overall Macro-F1 and Preference accuracy for `Qwen3-4B`, achieving $+2.3$ boost on Macro-F1 and $+1.7$ on preference accuracy over the *binary only* reward.

## 4.4 Per-Dimension Performance and Single vs. Multi-Task

Next, we compare pairwise RL-trained `Qwen3-4B` models trained jointly across all dimensions versus individually per dimension. Across the four dimensions, *Tone* achieves the strongest performance, with a Macro-F1 of 43.1 in the zero-shot setting and gains of up to $+3.8$ with multitask RL. *Localized Tone* and *Fluency* follow, reaching 32.8 and 32.2 Macro-F1 in zero-shot, and up to $+5.7$ improvement when trained with multitask RL. In constrast, *Localized Factuality* lags behind the other dimensions, achieving only 22.5 Macro-F1 in zero-shot, a trend consistent across all models. Moreover, RL yields limited benefit ($+0.6$ in single-task RL) or even regressions in the multitask setup. These results highlight the challenge of localized factuality and suggest that alternative strategies, such as incorporating retrieval, search, or external tool use, may be necessary. Full results are provided in Table 20 in Appendix E.

Overall, aside from *Localized Factuality*, joint multi-dimension training performs on par with single-dimension optimization while offering greater efficiency and practical benefits, such as serving as a reward model for post-training, which we explore in §5.

## 4.5 Cross-Language Performance

Figure 3 shows Preference accuracy per language variety for RL-trained `Qwen3-4B`. Performance varies widely, with `tr_TR` at $82.1\%$ and `bn_BD` at $37.9\%$, and does not strictly align with high- vs.

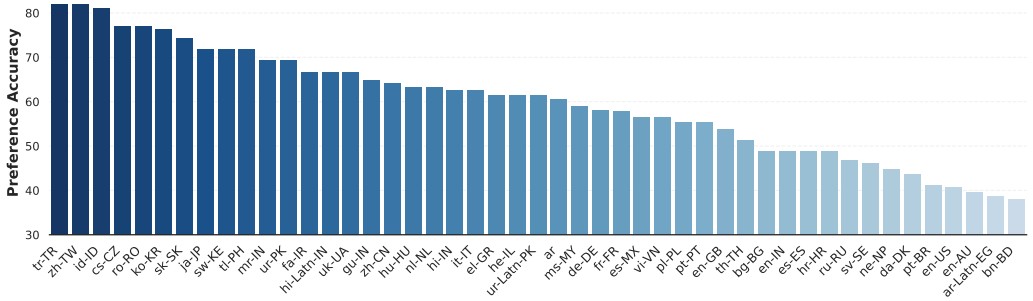

Figure 3: Preference Accuracy per Language of *pairwise* RL-trained `Qwen3-4B`.

low-resource languages. Relative to the zero-shot baseline, `en_AU` and `fr_FR` achieve the largest Macro-F1 gains ($+20.9\%$, $+17.7\%$) and `ro_RO` and `gu_IN` the largest Preference accuracy gains ($+18.0\%$, $+16.2\%$). By contrast, `es_ES` drops $-15.4\%$ in Preference despite a modest $+2.2\%$ Macro-F1 gain, whereas `en_MX`, the same language but a different locale, sees $+2.6\%$ and $+9.8\%$ gains in Preference and Macro-F1 , highlighting that our dataset captures language variety nuances.

We further trained RL using only English data and evaluated on all languages. Performance degrades compared to the baseline, indicating that English-only training is insufficient to generalize across all 47 language varieties. Detailed per-language variety performance is provided in Appendix E.2.

# 5 FROM LLM-JUDGES TO REWARD MODELS

We next investigate whether the RL-trained pairwise judges developed in our framework can also serve as generative reward models to directly improve LLM native-like response quality, unifying evaluation and optimization in a single framework.

## 5.1 RL WITH JUDGES AS GENERATIVE REWARD MODELS

For efficiency, we focus on smaller models for these experiments: `Qwen3-4B` as the policy model, and `Qwen3-4B-RL-Judge` as the reward model (RM). Since Localized Factuality remains challenging for our judges, we restrict both training and evaluation to Fluency, Tone, and Localized Tone. Specifically, we exclude Localized Factuality from all training and test prompts, randomly sample 3,000 prompts from MENLO for training, and retain all 1,398 test prompts from MENLO for evaluation across the three selected dimensions.

We post-train `Qwen3-4B` with GRPO. We sample 8 rollouts per prompt and compute rewards as follows: for each prompt, we construct response pairs from the rollouts, format them with the same pairwise evaluation template, and feed them to the RM. The final reward of each rollout is obtained by averaging its scores across all paired comparisons. Training details are added in Appendix D.2.

## 5.2 TWO-STAGE EVALUATION STRATEGY

To rigorously evaluate the policy model's native-like quality improvements, we employ a two-stage validation approach: (i) comprehensive automated evaluation across all 47 language varieties using three diverse LLM judges, and (ii) human validation on a strategically selected subset of 10 high-resource languages where we can ensure annotation quality.

For each test prompt, we generate responses from both the baseline (`Qwen3-4B`) and post-trained (`Post-train`) models, construct response pairs with randomized order to mitigate positional bias, and apply the same pairwise judge template used in training.

**LLM-Judges Evaluation** We select three high-performing judges (see §3) `Qwen3-32B`, `gpt-4.1`, and `Llama4-Scout-RL-Judge`, and compute win, loss, and tie rates between baseline and post-trained models, along with average scores on a 1–5 scale across all 1,398 test prompts spanning 47 language varieties. `Qwen3-4B-RL-Judge` is excluded from evaluation to avoid potential bias, since

Table 7: Two-stage Evaluation of `Qwen3-4B` and its RL post-trained variant `Post-train` on the MENLO test set, where both models serve as *response models*.

| JUDGES/RATERS | # Languages | Win Rate | | | Average Score (1–5) | | | |
|---|---|---|---|---|---|---|---|---|
| | | Post-train Win | Post-train Loss | Tie | Qwen3-4B | Post-train | $\Delta Sore$ | Improvement% |
| `Llama4-Scout-RL-Judge` | 47 | 63.88% | 9.16% | 26.96% | 3.01 | 3.79 | +0.78 | +25.9% |
| `Qwen3-32B` | 47 | 72.46% | 21.89% | 6.65% | 3.44 | 4.29 | +0.85 | +24.7% |
| `gpt-4.1` | 47 | 77.90% | 11.30% | 10.80% | 3.21 | 4.37 | +1.16 | +36.1% |
| `Llama4-Scout-RL-Judge` | 10 | 69.66% | 9.64 % | 20.70% | 3.36 | 4.22 | +0.86 | +25.6% |
| **Human Raters** | 10 | 55.71% | 35.20% | 9.09% | 3.31 | 3.67 | +0.36 | +10.9% |

it serves as the RM. Table 7 shows that the post-trained policy model consistently outperforms the baseline across all LLM judges and languages. Average score improvements range from $+0.80$ to $+1.16$, with win rates between 63.4% and 77.9%. Per-dimension analysis reveals consistent gains across evaluation criteria: Tone yields the largest improvement ($+1.04$ average score boost), followed by Localized Tone and Fluency ($+0.89$ each). The consistency of improvements across different judge architectures and all three dimensions provides strong evidence for the effectiveness of our reward modeling approach.

**Human Validation** To anchor our automated evaluation results, we conduct human evaluation on a diverse subset of 10 higher-resource languages: `ar`, `de_DE`, `en_US`, `fr_FR`, `hi_IN`, `hi_Latn_IN`, `pt_BR`, `tl_PH`, `th_TH`, `vi_VN`. This subset spans multiple language families, scripts, and geographic regions while ensuring access to qualified native speaker annotators. Human evaluation follows the same pairwise annotation guidelines as in MENLO construction.

Results (last row of Table 7) on the subset confirms the automated evaluation trends. The post-trained model achieves a win rate of $55.7\%$ against the baseline, with an average score improvement of $+10.9\%$. While both automated and human evaluators agree that post-training improves response quality, we observe that LLM judges tend to overestimate the magnitude of improvement compared to human raters. Comparing human evaluations to the closest-performing automated judge (`Llama4-Scout-RL-Judge`) on this subset reveals systematic differences: the automated judge reports an average improvement of $+0.5$ higher than humans. We hypothesize that this discrepancy arises because the automated judges may lean towards a stylistic caricature of native-like quality, overestimating improvements relative to nuanced human judgments. In addition, RL-trained judges exhibit less of this discrepancy among LLM evaluators, confirming the benefits of our RL judge training.

Overall, our two-stage evaluation demonstrates the potential of RL-trained judges as generative reward models for aligning multilingual outputs toward native-like quality. The directional consistency observed across both LLM- and human-based evaluations validates the viability of our unified framework for multilingual proficiency alignment. However, we note that challenges remain: LLM judges tend to overestimate the magnitude of improvements relative to human raters, highlighting an important direction for future work.

## 6 RELATED WORK

**Multilingual Evaluation** Models' multilingual proficiency has been typically measured as an aggregate of performance across multiple task-oriented evaluations of short-form responses in settings with verifiable answers (Hu et al., 2020; Ruder et al., 2021; Doddapaneni et al., 2023; Ahuja et al., 2023; 2024). Recent benchmarks focused on the evaluation of model's cultural knowledge in a similar verifiable setting (Myung et al., 2024; Chiu et al., 2025; Fabbri et al., 2025). However, such evaluations do not extend to real-world conversations containing long-form responses. Benchmarks evaluating long-form responses use prompts and responses translated from English (Son et al., 2024; Liu et al., 2024; Doddapaneni et al., 2025; Gureja et al., 2025). These evaluations typically do not reflect more localized aspects of language quality and are biased towards translationese. Son et al. (2024) and Doddapaneni et al. (2025) automatically generate 'good' and 'bad' responses for each dimension. Marchisio et al. (2024) and Guo et al. (2025) evaluate language consistency and naturalness respectively in relatively narrow settings. PARIKSHA (Watts et al., 2024) is the most similar dataset to ours as it uses human-written prompts and human-annotated responses, but focuses on 10

Indic languages, annotates only high-level dimensions, and reports moderate inter-annotator agreement. MENLO is the only dataset that focuses on native-like quality in real-world conversations.

**Multilingual Judges and RMs** LLMs have been used as judges in different multilingual benchmarks (Liu et al., 2024; Fabbri et al., 2025). However, fewer works focus on analyzing or improving multilingual judges and RMs. Gureja et al. (2025) observe that zero-shot judges show a substantial gap between the translated M-REWARDBENCH and its English counterpart, with predictions inconsistent across languages. Fu & Liu (2025) report similar inconsistencies across five diverse tasks. Wu et al. (2024) evaluate zero-shot cross-lingual transfer of trained RMs on summarization and dialog, observing gains. Hong et al. (2025) find strong cross-lingual transfer on M-REWARDBENCH for English RMs fine-tuned in four languages. Doddapaneni et al. (2025) fine-tune a judge with SFT on automatically translated prompts and responses in six languages to produce an absolute score. To our knowledge, we are the *first* to (i) train judges and RMs in a massively multilingual setting, (ii) fine-tune multilingual judges with RL, and (iii) demonstrate the benefits of multi-task RL, reward shaping, and pairwise grading in this setting.

## 7 CONCLUSION

We introduce MENLO, a comprehensive framework for evaluating and improving native-like response quality across 47 language varieties. By combining sociolinguistically-informed prompt design, detailed evaluation rubrics, and high-quality human annotations, MENLO captures multiple dimensions of conversational proficiency, including fluency, tone, localized tone, and localized factuality. We demonstrate that pairwise evaluation significantly improves both zero-shot and fine-tuned LLM judges, and that RL with reward shaping yields best judge performance.

Beyond evaluation, we show these trained judges can serve as generative reward models to directly improve policy model's response quality. While challenges remain with the tendency of LLM judges to overestimate improvements relative to human raters, our framework provides a practical and scalable approach to both assessing and enhancing LLM proficiency in multilingual context.

### ACKNOWLEDGMENTS

We would like to thank Paco Guzmán for guidance on the original direction of this project. We would like to thank Pritish Yuvraj for help with initial generations. We thank Max Mauer and Nidhi Nisarg Shah for support on the annotation interface. We would like to thank Wes Kranz, Lora Zhou, and Kish Patel for help on operations. We thank the Multilingual Post-training team for discussions and feedback.

### ETHICS STATEMENT

Translators and annotators were recruited through third-party services and compensated based on local regulations.

### REPRODUCIBILITY STATEMENT

We release the full MENLO to the community. All models are built on top of open-weight Llama and Qwen backbones. The prompt templates used for training are provided in Appendix C, and detailed descriptions of experimental setups, hyperparameters, and libraries are included in Appendix D.

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

# APPENDIX

## A  ADDITIONAL DETAILS ON THE MENLO DATASET

### A.1  DATASET COLLECTION

**Localization vs natively written prompts**   Our prompts derive from English templates to ensure coverage consistency across a large number of languages. While localization includes cultural adaptation, we acknowledge that independently authored prompts in each language would better capture native discourse structures. We focus our resources on native speakers annotating responses instead.

We initiated a pilot to evaluate of different models including GPT-4o and Llama4-Maverick in a single category, Localized Tone focusing on five languages spoken by the authors: Bengali, German, Hindi, Italian, and Russian. As expected, the pilot framework was quickly confronted with the complexities inherent in multilingualism: Even among the authors, for all initial Localized Tone prompts, we struggled to reach consistent and reliable agreement. Nevertheless, these early results provided valuable insights to guide us to improve prompt design, guideline clarity, and annotation arrangement.

To enhance the reliability of the framework, we took steps to refine our prompts' nuance and complexity, update guidelines with clearer direction for annotators that aimed to make abstract concepts more concrete, and started exploring a more user-friendly annotation solution. The changes brought upon notable improvements in inter-annotator agreement that extended to the full-scale annotation. We show the agreement of the initial pilot annotation and improved annotation for localized tone in 5 languages in Table 9 and across all categories and languages in Table 9.

Table 8: Comparison of PILOT and MENLO for 5 languages in localized tone category. *Agreement is defined as the percentage of annotation pairs whose ratings for the same item differ by no more than 1.*

| Language Code | PILOT Agreement | MENLO Agreement |
|---|---|---|
| bn_BD | 0.75 | 0.84 |
| de_DE | 0.74 | 0.92 |
| hi_IN | 0.74 | 0.92 |
| it_IT | 0.79 | 0.71 |
| ru_RU | 0.71 | 0.79 |
| Overall | 0.75 | 0.84 |

Table 9: Comparison of the pilot annotation (5 languages) and final MENLO dataset (47 languages). *Agreement is defined as the percentage of annotation pairs whose ratings for the same item differ by no more than 1. Agreement for PILOT has been averaged over 5 languages, while MENLO is averaged over 47.*

| Quality Dimension | PILOT (5 Languages) | | | MENLO (47 Languages) | | |
|---|---|---|---|---|---|---|
| | Agreement | # Prompts | # Annotations | Agreement | # Prompts | # Annotations |
| Fluency | 0.76 | 150 | 450 | 0.82 | 1,820 | 23,556 |
| Tone | 0.70 | 150 | 450 | 0.77 | 1,410 | 18,712 |
| Localized Tone | 0.75 | 200 | 600 | 0.82 | 1,825 | 22,324 |
| Localized Factuality | 0.78 | 150 | 450 | 0.78 | 1,378 | 16,422 |
| Overall | 0.75 | 650 | 1,950 | 0.80 | 6,423 | 81,014 |

Table 10: Components to consider when annotating different subcategories of Tone.

| Tone Subcategory | Tone Component 1 | Tone Component 2 |
|---|---|---|
| Helpful Tone | Instruction following ✓ | Emotional support ✓ |
| Insightful Tone | Informative ✓ | Empathetic ✓ |
| Engaging Tone | Conversational Language ✓ | Encourages Interactions ✓ |
| Fair Tone | Non-biased stance ✓ | Non-preachy language ✓ |

**Annotation guidelines** Judging language performance can be subjective. To minimize confusion, we identified the most important components of tone, fluency, localized tone, and localized factuality and incorporated them into the guidelines. For example, a model response that conveys a helpful tone must succeed on two fronts: providing (or attempting to provide) help based on users' instructions, and expressing emotional engagement to sound caring. By breaking down broad linguistic concepts into easy-to-follow subcategories and self-explanatory signals (illustrated via emoji ✓ ⓞⓚ ✗ ?), annotators can quickly grasp and refer back to the guidelines. We show subcategories for Tone, for example, in Table 10 and show the rubric guidelines for Localized Factuality in Table 11.

**Annotation tool** To streamline the annotation, we developed a custom annotation interface, which we show in Figure 4. The tool provides a simple, annotator-friendly user interface for guidelines, rating model responses, and randomized model A/B pairwise comparison. The backend allowed us to ensure data is consistent and identify any missing annotations or other data-related issues. In addition, it enabled us to quickly test annotators on dedicated test annotations before moving them to the actual annotation tasks. Overall, solid tooling allowed us to screen more than 1,000 annotators and collect more than 80,000 annotations.

Table 11: Localized factuality rubrics for annotation using a 5-point Likert scale. Each rating corresponds to a high-level classification of the response (e.g., *"Sounds somewhat accurate and relevant"*), further specified by dimension-specific criteria e.g., accuracy, relevance, and completeness.

| 1: Major failure *"Grossly incorrect or misleading"* | 2: Minor failure *"Some mistakes"* | 3: Pass *"Sounds somewhat accurate and relevant"* | 4: Good *"Sounds accurate and relevant"* | 5: Excellent *"Factually accurate, highly relevant, complete with additional info"* |
|---|---|---|---|---|
| **Accuracy** ✗✗ 
 - The model's response contains obvious factual errors or made-up information. | **Accuracy** ✗? 
 - The model's response contains some factual mistakes. | **Accuracy** (OK) 
 - No obvious factual errors but some claims are not entirely correct or may be misleading. | **Accuracy** ✓ 
 The claims in the response are factually accurate. | **Accuracy** ✓✓ 
 The claims in the response are completely factually accurate. |
| **Locale Relevance** ✗✗ 
 **Local Point of View** ✗✗ 
 - The model fails to understand the basic local context. 
 - The model provides content that is irrelevant or misaligned with the local context. 
 - The model's response frames the answer in a fetishizing/offensive way (like overly explaining basic local knowledge to locals) | **Locale Relevance** ✗? 
 **Local Point of View** ✗? 
 - The model grasps some local context but misses key nuances. 
 - The model's response is somewhat relevant but provides mainly general or high-level information that lacks alignment with the local context. 
 - The model's response may come across as slightly insensitive or tone-deaf, but it does not contain overtly fetishizing or offensive answers. | **Locale Relevance** (OK) 
 **Local Point of View** (OK) 
 - The model generally understands the local context but may miss subtle nuances. 
 - The response is generally relevant and aligned with the local context. 
 - The model's response is neutral and factual. | **Locale Relevance** ✓ 
 **Local Point of View** ✓ 
 - The model accurately interprets the local context and nuances. 
 - The response is generally relevant and aligned with the local context. 
 - The model avoids explanations that might be seen as overly simplistic or patronizing. Instead, the facts are thoughtfully selected with depth. | **Locale Relevance** ✓✓ 
 **Local Point of View** ✓✓ 
 - The model demonstrates a deep understanding of the local context and nuances. The response delivers highly relevant content that is highly specific and perfectly aligned with the local context. 
 - The model chooses facts that are in-depth and nuanced even for someone who's already a local. It might present additional context and highlights regional variations to show depth of local knowledge. |
| **Completeness** ✗✗ 
 - The model's response is incomplete and misses crucial information to answer the question. | **Completeness** ✗? 
 - The response answers part of the question but is missing some relevant pieces of information. | **Completeness** (OK) 
 - The model provides sufficient information to answer the question but the provided information may lack depth. | **Completeness** ✓ 
 - The model provides all the information to answer the question. | **Completeness** ✓✓ 
 - The response is rich in information and covers all information to answer the question as well as additional helpful context that further helps to contextualize the response. |

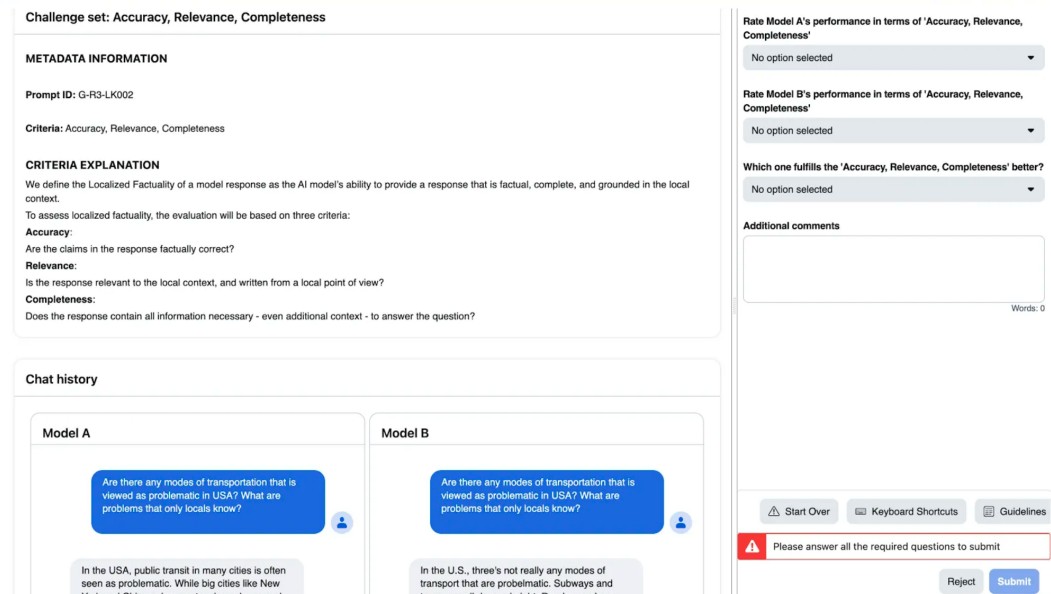

Figure 4: Annotation interface used for MENLO.

## A.2 LANGUAGE VARIETIES IN MENLO

MENLO covers 47 language varieties. Table 12 lists each variety along with its corresponding ISO 639-1 code.

Table 13 reports annotator IAA by dimension and language variety.

Table 12: Mapping from language-region codes to language names.

| Language code | Full name | Language code | Full name |
|---|---|---|---|
| ar | Modern Standard Arabic | mr_IN | Marathi |
| ar_Latn_EG | romanized Egyptian Arabic | ms_MY | Malay (Malaysia) |
| bg_BG | Bulgarian | ne_NP | Nepali |
| bn_BD | Bengali | nl_NL | Dutch |
| cs_CZ | Czech | pl_PL | Polish |
| da_DK | Danish | pt_BR | Brazilian Portuguese |
| de_DE | German | pt_PT | Portuguese (Portugal) |
| el_GR | Greek | ro_RO | Romanian |
| en_AU | Australian English | ru_RU | Russian |
| en_GB | British English | sk_SK | Slovak |
| en_IN | Indian English | sv_SE | Swedish |
| en_US | US English | sw_KE | Swahili (Kenya) |
| es_ES | Spanish (Spain) | th_TH | Thai |
| es_MX | Mexican Spanish | tl_PH | Tagalog (Philippines) |
| fa_IR | Persian (Iran) | tr_TR | Turkish |
| fr_FR | French (France) | uk_UA | Ukrainian |
| gu_IN | Gujarati (India) | ur_Latn_PK | romanized Urdu |
| he_IL | Hebrew (Israel) | ur_PK | Urdu |
| hi_IN | Hindi | vi_VN | Vietnamese |
| hi_Latn_IN | romanized Hindi | zh_CN | Chinese (China) |
| hr_HR | Croatian | zh_TW | Traditional Chinese (Taiwan) |
| hu_HU | Hungarian | ja_JP | Japanese |
| id_ID | Indonesian | ko_KR | Korean |
| it_IT | Italian | | |

Table 13: Krippendorff alpha by quality dimension and language.

| Language Code | Tone | Fluency | Localized Tone | Localized Factuality | Average |
|---|---|---|---|---|---|
| ar | 0.83 | 0.79 | 0.86 | 0.79 | 0.82 |
| ar_Latn_EG | 0.86 | 0.79 | 0.82 | NA | 0.82 |
| bg_BG | 0.75 | 0.80 | 0.79 | 0.86 | 0.80 |
| bn_BD | 0.82 | 0.79 | 0.85 | 0.80 | 0.82 |
| cs_CZ | 0.80 | 0.78 | 0.78 | 0.82 | 0.79 |
| da_DK | 0.83 | 0.78 | 0.78 | 0.85 | 0.81 |
| de_DE | 0.82 | 0.76 | 0.85 | 0.77 | 0.80 |
| el_GR | 0.83 | 0.85 | 0.85 | 0.85 | 0.84 |
| en_AU | 0.89 | 0.73 | 0.81 | 0.82 | 0.81 |
| en_GB | 0.85 | 0.79 | 0.85 | 0.82 | 0.83 |
| en_IN | 0.84 | 0.85 | 0.81 | 0.83 | 0.83 |
| es_ES | 0.78 | 0.78 | 0.81 | 0.79 | 0.79 |
| es_MX | 0.79 | 0.80 | 0.86 | 0.84 | 0.82 |
| fa_IR | 0.83 | 0.82 | 0.77 | 0.81 | 0.81 |
| fr_FR | 0.83 | 0.76 | 0.81 | 0.82 | 0.81 |
| gu_IN | 0.89 | 0.81 | 0.86 | 0.84 | 0.85 |
| he_IL | 0.85 | 0.79 | 0.80 | 0.84 | 0.82 |
| hi_IN | 0.82 | 0.81 | 0.83 | 0.87 | 0.83 |
| hi_Latn_IN | 0.86 | 0.77 | 0.83 | 0.80 | 0.82 |
| hr_HR | 0.81 | 0.78 | 0.80 | 0.82 | 0.80 |
| hu_HU | 0.85 | 0.81 | 0.80 | 0.84 | 0.82 |
| id_ID | 0.90 | 0.81 | 0.82 | 0.82 | 0.84 |
| it_IT | 0.83 | 0.77 | 0.77 | 0.85 | 0.80 |
| ja_JP | 0.86 | 0.82 | 0.79 | 0.81 | 0.82 |
| ko_KR | 0.86 | 0.83 | 0.80 | 0.84 | 0.83 |
| mr_IN | 0.88 | 0.78 | 0.78 | 0.86 | 0.82 |
| ms_MY | 0.84 | 0.82 | 0.81 | 0.83 | 0.83 |
| ne_NP | 0.83 | 0.79 | 0.80 | 0.83 | 0.81 |
| nl_NL | 0.84 | 0.81 | 0.84 | 0.79 | 0.82 |
| pl_PL | 0.83 | 0.79 | 0.86 | 0.82 | 0.83 |
| pt_BR | 0.86 | 0.82 | 0.82 | 0.85 | 0.83 |
| pt_PT | 0.83 | 0.80 | 0.83 | 0.80 | 0.82 |
| ro_RO | 0.84 | 0.79 | 0.80 | 0.82 | 0.81 |
| ru_RU | 0.81 | 0.75 | 0.80 | 0.78 | 0.79 |
| sk_SK | 0.88 | 0.81 | 0.81 | 0.84 | 0.83 |
| sv_SE | 0.84 | 0.78 | 0.81 | 0.81 | 0.81 |
| sw_KE | 0.88 | 0.84 | 0.83 | 0.85 | 0.85 |
| th_TH | 0.85 | 0.83 | 0.78 | 0.83 | 0.82 |
| tl_PH | 0.84 | 0.83 | 0.82 | 0.81 | 0.83 |
| tr_TR | 0.88 | 0.85 | 0.79 | 0.80 | 0.83 |
| uk_UA | 0.89 | 0.77 | 0.81 | 0.80 | 0.82 |
| ur_Latn_PK | 0.82 | 0.79 | 0.80 | 0.86 | 0.82 |
| ur_PK | 0.81 | 0.79 | 0.82 | 0.86 | 0.82 |
| vi_VN | 0.85 | 0.82 | 0.82 | 0.82 | 0.83 |
| zh_CN | 0.86 | 0.81 | 0.80 | 0.83 | 0.83 |
| zh_TW | 0.89 | 0.82 | 0.80 | 0.89 | 0.85 |

## A.3 GRADING RUBRICS

The 5-point grading rubrics are defined for each question type under the four dimensions:

**Fluency**: *Vocabulary & Syntax*, *Coherence*, *Grammar & Mechanics*, *Clarity & Conciseness*.

**Localized Tone**: *Cultural Relevance*, *Formality & politeness*, *Humor*, *Linguistic nuance*.

**Localized Factuality**: *Cultural Practices*, *Expressions & Concepts*, *Local Knowledge*.

**Tone**: *Be engaging*, *Be fair*, *Be insightful*, *Help as best as you can*.

The rubrics were created based on reviews of example prompts and failure modes of the different dimensions and inspired by prior work on automated proficiency assessment (Ke & Ng, 2019) and cross-cultural variation (Hershcovich et al., 2022; Myung et al., 2024).

All rubrics use the same 5-point scale, with criteria adapted to the specific question type. We show some examples of the grading rubrics in Figure 5, 6, 7, and 8.

---

**Grading Rubrics for Localized Tone**

```
### Grading Criteria:

1 - MAJOR FAILURE
* The response shows no understanding of formal or informal language, or uses an
overly formal/informal tone that is not suited to the context.

2 - MINOR FAILURE
* The response shows limited understanding of formal or informal language, with
significant errors or misunderstandings.

3 - PASS
* The response does not contain any significant formality errors but also does not
use the most appropriate formality or politeness markers or formulations.

4 - GOOD
* The response shows good use of formal or informal language. Also appropriate
formality/formatting for the task, such as letter, application form, etc.

5 - EXCELLENT
* The response shows excellent use of formal or informal language, with a tone that
is perfectly suited to the context.
* The response shows excellent local formality/formatting for the task, such as
letter, application form, etc.
```

Figure 5: Example of 5-Point Grading Rubrics for **Localized Tone** (*Formality & politeness*).

Grading Rubrics for **Fluency**

```
### Grading Criteria:

1 - MAJOR FAILURE
* The response is full of mistakes and hard to understand.
* The response lacks a clear structure or logical flow.
* Ideas are disconnected or jump abruptly from one topic to another.
* The response contains numerous grammatical errors.
* The response contains numerous punctuation or capitalization errors, or typos.
* The response frequently misuses words | out of context, or improper regional
variants (e.g., lift/elevator).
* Sentence structure is awkward or repetitive.
* The response is unclear or convoluted.
* Ideas are expressed in a roundabout or overly verbose manner.

2 - MINOR FAILURE
* Parts of the response are vaguely understandable.
* Some logic connections are not clear.
* Some topics are loosely connected.
* Transitions feel forced or abrupt.
* The response contains some grammatical errors.
* The response contains some punctuation and capitalization errors.
* The response contains some awkwardness or repetitiveness.
* Some sentences are difficult to understand due to unclear language.
* Some parts are overly verbose.

3 - PASS
* The response is understandable.
* Text is somewhat coherent and understandable.
* Merits may balance out failures.
* The response contains no major grammatical errors, but is also not outstanding
in writing.
* The response contains no major flaws in word choices and syntax, but lacks
nuances and sophistication.
* Sentence structure is plain or basic.
* Language is generally clear and of appropriate length.

4 - GOOD
* The response is easily understandable.
* Ideas are connected and fluency is good.
* The response is grammatically correct and free of errors.
* Words are used accurately and in context.
* Sentence structure varies, with a mix of simple, compound, and complex sentences.
* The text is easy to understand, with no unnecessary words or phrases.
* Ideas are expressed clearly and directly, with the use of advanced structures
such as bullet points.

5 - EXCELLENT
* The response is fluent and natural.
* The text is well-organized and logically structured.
* Ideas are connected and flow smoothly.
* The response is free of grammatical errors. Complex sentences are constructed
thoughtfully, avoiding run-ons or awkward phrasing.
* Correct and sophisticated use of tense, punctuations (question marks, exclamation
marks, etc.).
* Great word choices that enhance clarity and depth.
* Great variety of different types of sentences, including simple, compound and
complex sentences.
* The text is effortlessly comprehensible, with no ambiguity or confusion, and every
word serves a purpose.
* Ideas are conveyed directly, without redundancy or verbosity, ensuring maximum
impact with minimal words.
* The response effectively uses bullet points and other methods to enhance clarity.
```

Figure 6: Example of 5-Point Grading Rubrics for **Fluency**.

Grading Rubrics for **Tone**

```
### Grading Criteria:

1 - MAJOR FAILURE
* The response provides no insights whatsoever.
* The response refuses to provide help when it should.
* The response fails to deliver the right information, resulting in incomplete or
inaccurate answers.
* The response fails to explain why things matter, lacking a clear chain of thought.
* The response does not exhibit intellectual curiosity towards the user or the
subject. Lack of interest or engagement.
* The response does not show empathy towards the user. The response is offensive
and distant.

2 - MINOR FAILURE
* The response ATTEMPTS to be insightful but lacks accuracy or substance.
* The response provides some relevant information but lacks completeness or
accuracy.
* The response provides some explanations but lacks clarity or relevance.
* The response shows some curiosity but lacks depth or engagement.
* The response asks some follow-up questions but lacks specificity or relevance.

3 - PASS
* The response provides some insights but feels dry.
* The response generally is informative but may miss some nuances.
* The response uses facts to draw sound conclusions and connections.
* The response generally delivers complete and accurate information but may miss
some details.
* The response generally exhibits intellectual curiosity but may lack consistency.
* The response asks follow-up questions but may lack tailoring to the user's needs.
* The response provides the information the user is seeking but may lack depth.

4 - GOOD
* The response sounds insightful and knowledgeable.
* The response clearly communicates the subject matter.
* The response effectively uses facts and examples to draw insightful conclusions
and connections.
* The response delivers complete and accurate information, fully answering the
user's query.
* The response utilizes unique tools (e.g. Reels plugin) when they are relevant to
the user's query.
* The response demonstrates curiosity, exploring the user's question with interest
and depth.
* The response asks tailored follow-up questions, enhancing the conversation.
* The response provides the information the user is seeking, meeting their needs.

5 - EXCELLENT
* The response is insightful with shrewd observation and critical perspective,
making the user aware of nuances about the subject.
* The response clearly communicates the subject matter.
* The response effectively uses facts and examples to draw insightful conclusions
and connections.
* The response uses critical thinking and various perspectives to provide in-depth,
comprehensive information that exceeds user expectations.
* The response utilizes unique tools (e.g. Reels plugin) when they are relevant to
the user's query.
* The response provides thoughtful observation into the user's prompt.
* The response asks insightful and tailored follow-up questions, deeply engaging
with the user's needs.
* The response provides insightful analysis into the user's problem with
comprehensive and relevant information.
```

Figure 7: Example of 5-Point Grading Rubrics for **Tone** (*Be insightful:Be intellectually curious and engaging*).

---

Grading Rubrics for **Localized Factuality**

```
### Grading Criteria:

1 - MAJOR FAILURE
* The response is grossly incorrect or misleading.
* The response contains obvious factual errors or made-up information.
* The response fails to understand the basic local context.
* The response provides content that is irrelevant or misaligned with the local
context.
* The response frames the answer in a fetishizing/offensive way (like overly
explaining basic local knowledge to locals.)
* The response is incomplete and misses crucial information to answer the question.

2 - MINOR FAILURE
* The response contains some mistakes.
* The response contains some factual mistakes.
* The response grasps some local context but misses key nuances.
* The response is somewhat relevant but provides mainly general or high-level
information that lacks alignment with the local context.
* The response may come across as slightly insensitive or tone-deaf, but it does
not contain overtly fetishizing or offensive answers.
* The response answers part of the question but is missing some relevant pieces of
information.

3 - PASS
* The response sounds somewhat accurate and relevant.
* There are no obvious factual errors but some of the claims are not entirely
correct or may be misleading.
* The response generally understands the local context but may miss subtle nuances.
* The response is generally relevant and aligned with the local context.
* The response is neutral and factual.
* The response provides sufficient information to answer the question but the
provided information may lack depth.

4 - GOOD
* The response sounds accurate and relevant.
* The claims in the response are factually accurate.
* The response accurately interprets the local context and nuances.
* The response is generally relevant and aligned with the local context.
* The response avoids explanations that might be seen as overly simplistic or
patronizing. Instead, the facts are thoughtfully selected with depth.
* The response provides all the information to answer the question.

5 - EXCELLENT
* The response is factually accurate, highly relevant, and complete with additional
information.
* The claims in the response are completely factually accurate.
* The response demonstrates a deep understanding of the local context and nuances.
The response delivers highly relevant content that is highly specific and perfectly
aligned with the local context.
* The response chooses facts that are in-depth and nuanced even for someone who's
already a local.  It might present additional context and highlights regional
variations to show depth of local knowledge.
* The response is rich in information and covers all information to answer the
question as well as additional helpful context that further helps to contextualize
the response.
```

Figure 8: Example of 5-Point Grading Rubrics for **Localized Factuality**.

A.4   FULL EXAMPLES FOR MENLO

We provide full examples from MENLO in Figure 9, 10, 11, 12, 13, 14, 15, and 16, including prompt (both in English and the translated version in target languages), responses, and corresponding grades. Examples cover different languages and dimensions.

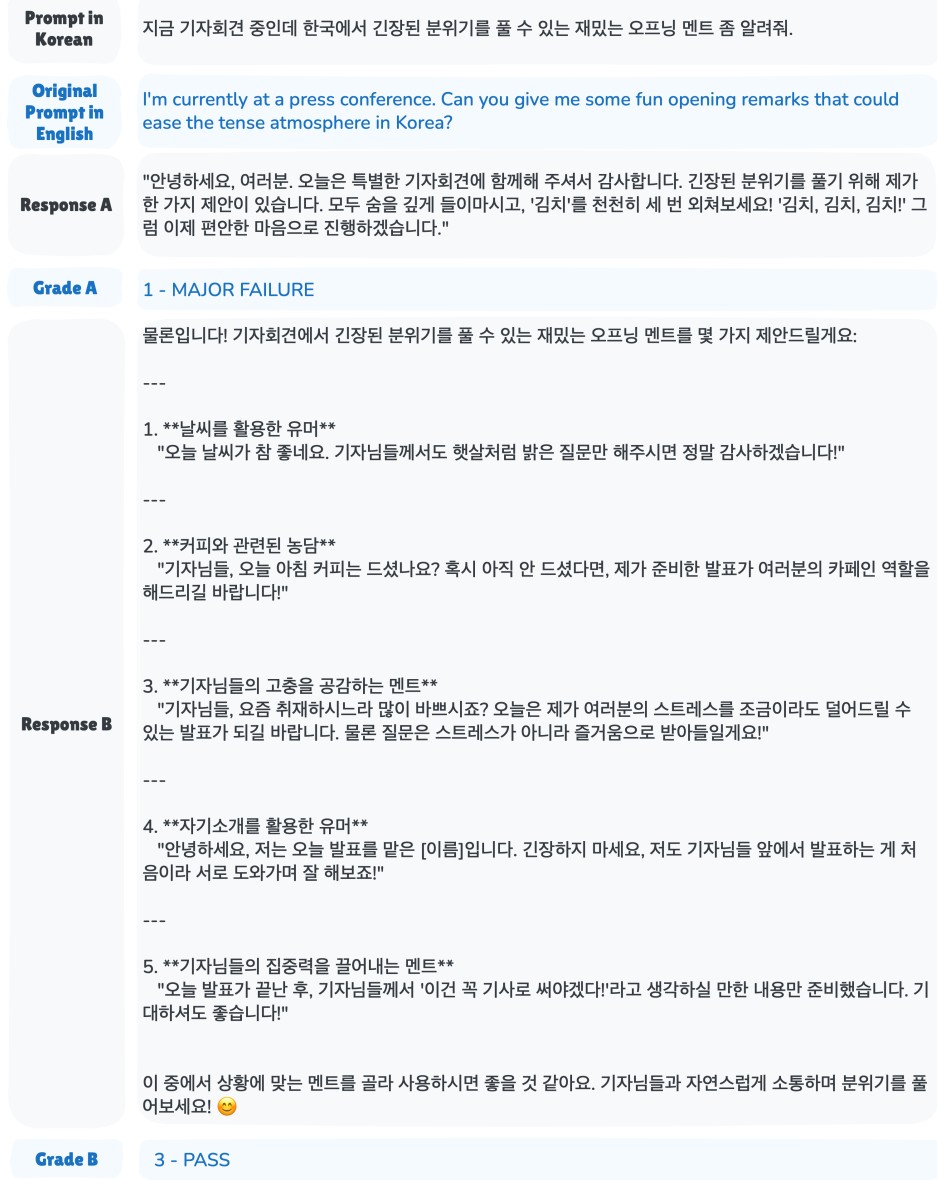

Figure 9: Example prompt, responses, and annotation in **Korean** for **Localized Tone** (Humor).

| **Prompt in Czech** | Napiš chytlavou obchodní prezentaci pro investory o revolučním místním hotelovém byznusu v Česku, kterou jde říct za míň než 3 minuty |
|---|---|
| **Original Prompt in English** | Write a catchy business presentation for investors about the revolutionary local hotel business in the Czech Republic, which can be said in less than 3 minutes |

**Response A**

**Prezentace: Revoluce v českém hotelovém byznysu**

Dobrý den, vážení investoři,
představte si, že cestujete po Česku. Hledáte ubytování, které není jen postel a střecha nad hlavou, ale zážitek. Místo, kde se cítíte jako doma, kde poznáte místní kulturu, ochutnáte regionální kuchyni a kde vás přivítají jako člena rodiny. Přesně to přináší náš revoluční hotelový koncept.

Naše hotely nejsou jen ubytovací zařízení – jsou to živé komunitní centra. Spojujeme moderní technologie s autentickým českým zážitkem. Každý hotel je navržen tak, aby odrážel unikátní charakter regionu, od designu interiéru až po nabídku aktivit. Hosté si mohou vychutnat lokální produkty, zapojit se do workshopů s místními řemeslníky nebo se vydat na výlety s průvodci, kteří znají každý kout.

A teď to nejlepší: díky našemu inovativnímu digitálnímu systému dokážeme optimalizovat provoz, snížit náklady a maximalizovat zážitek hostů. Naše aplikace umožňuje hostům snadno rezervovat, komunikovat s personálem, objednávat služby a dokonce sdílet své zážitky s ostatními.

Český turistický trh roste – jen v loňském roce vzrostl počet zahraničních turistů o 15 %. A přesto většina hotelů stále funguje podle zastaralých modelů. My přinášíme změnu. Naše první pilotní hotely již dosahují 90% obsazenosti a průměrné hodnocení 4,8 z 5.

Hledáme partnery, kteří s námi chtějí růst. Vaše investice nám umožní rozšířit tento koncept do dalších regionů, přilákat více turistů a vytvořit síť hotelů, která bude synonymem pro autentický český zážitek.

Pojďme společně změnit způsob, jakým lidé cestují po Česku. Přidejte se k nám a staňte se součástí revoluce v hotelovém byznysu. Děkuji!

**Grade A**  4 - GOOD

**Response B**

Dobrý den, vážení investoři,

Představujeme vám revoluční projekt v oblasti hotelového byznysu v Česku, který mění tradiční pojetí cestování a ubytování. Náš koncept se zaměřuje na propojení lokálních zážitků s moderním komfortem, vytvářejíc unikátní prostředí pro naše hosty.

V srdci České republiky vzniká síť hotelů, které nejsou jen místem k přespání, ale stávají se bránou do autentických lokálních zážitků. Naše hotely jsou navrženy tak, aby reflektovaly charakter destinace, ve které se nacházejí, od historické architektury po současnou kulturu a gastronomii.

Našimi klíčovými prvky jsou:
- **Lokální autenticita**: Každý hotel je designován ve spolupráci s místními umělci a řemeslníky, což zajišťuje jedinečný a neopakovatelný zážitek.
- **Udržitelný rozvoj**: Zaměřujeme se na ekologicky šetrné postupy a materiály, minimalizujíce náš dopad na životní prostředí.
- **Technologická inovace**: Implementace nejnovějších technologií pro zvýšení komfortu našich hostů, od chytrých systémů řízení teploty a osvětlení až po mobilní aplikace pro bezkontaktní check-in a personalizované služby.

Proč investovat do našeho projektu?
1. **Rostoucí poptávka po unikátních zážitcích**: Cestovatelé stále více hledají autentické a nezapomenutelné zážitky, které jdou nad rámec standardního cestovního ruchu.
2. **Neuspokojená potřeba lokálního hotelového trhu**: Tradiční hotely často postrádají osobitý charakter a dostatečně nevyužívají potenciál lokální kultury a tradic.
3. **Silný managementový tým**: Náš tým se skládá z odborníků s dlouholetými zkušenostmi v oblasti hotelového managementu, designu a udržitelného rozvoje.

Očekáváme vysoký růstový potenciál díky kombinaci inovativního přístupu, rostoucí poptávky po kvalitním a autentickém cestovním ruchu a pevnému závazku k udržitelným praktikám. Připojte se k nám a staňte se součástí revoluce v českém hotelovém průmyslu. Děkujeme za pozornost a těšíme se na možnost spolupráce.

**Grade B**  3 - PASS

Figure 10: Example prompt, responses, and annotation in **Czech** for **Localized Tone** (Cultural relevance).

**Prompt in Hebrew**

אני עומד להגיע לנמל התעופה בן גוריון בתל אביב ועליי להגיע להילטון תל אביב במרכז העיר באמצעות תחבורה ציבורית. לאחר איסוף הכבדה שלי, איזה אמצעי תחבורה ציבורית אני עשוי לראות קודם כשאני יוצא מנמל התעופה? האם יש אופציות תחבורה שמבקרים עשויים לפספס? מה האופציה הטובה ביותר עבור מישהו שנוסע עם מטען קל ואיזו מהן היא הטובה ביותר עבור מישהו הנושא שלוש מזוודות?

**Original Prompt in English**

I am about to arrive at Ben Gurion Airport in Tel Aviv and I have to get to the Tel Aviv Hilton in the city center by public transportation. After collecting my baggage, what form of public transport might I see first when I leave the airport? Are there transportation options that visitors might miss? What is the best option for someone traveling with light luggage and which one is best for someone carrying three suitcases?

**Response A**

לאחר שתאסוף את המזוודות שלך בנמל התעופה בן גוריון, תראה תחילה את תחנת האוטובוסים, הנמצאת במפלס 2 (קומת התחבורה הציבורית) בטרמינל 3. ישנם קווי אוטובוס רבים הפועלים משם, כולל קווים בינעירוניים ואזוריים. לעתים קרובות ניתן לראות גם תחנת מוניות קרוב לתחנות האוטובוס.

אופציה שבמקרים מסוימים מבקרים עשויים לפספס היא הרכבת. תחנת הרכבת נמצאת במרחק הליכה קצר מהטרמינלים, ומציעה אפשרות יעילה להגיע למרכז תל אביב.

**עבור נוסע עם מטען קל:** הרכבת היא האופציה הטובה ביותר. היא מהירה, נוחה ונוסעת ישירות לתחנת הרכבת תל אביב סבידור מרכז, קרוב למלון הילטון תל אביב.

**עבור נוסע עם שלוש מזוודות:** אוטובוס עשוי להיות פחות נוח בגלל המטען הרב. למרות שיש שירותי הסעה, העברת שלוש מזוודות לאוטובוס עשויה להיות קשה. מונית היא האופציה הנוחה ביותר במקרה זה, למרות שהיא תהיה יקרה יותר מהרכבת או האוטובוס.

שימו לב שזמני הנסיעה ותדירות התחבורה הציבורית עשויים להשתנות. מומלץ לבדוק את לוחות הזמנים העדכניים באמצעות אפליקציות כמו Moovit או גוגל מפות לפני הנסיעה.

**Grade A**

2 - Minor Failure

**Response B**

כשאתה יוצא מנמל התעופה בן גוריון בתל אביב, אמצעי התחבורה הציבורית הראשונים שתראה כסביר הם מוניות (טקסי) וקווי אוטובוסים.

קווי אוטובוסים כמו קו 445 או קווים אחרים של חברת אגד יכולים לקחת אותך למרכז תל אביב.

אופציות תחבורה שמבקרים עשויים לפספס לפספס כוללות את שירות ההסעות של נמל התעופה ליעדים ספציפיים במרכז העיר, אך לרוב מדובר בשירותים בתשלום ומראש.

האופציה הטובה ביותר עבור מישהו שנוסע עם מטען קל היא קו אוטובוס כמו קו 445, שהוא מהיר וזול יחסית.

האופציה הטובה ביותר עבור מישהו הנושא שלוש מזוודות היא מונית (טקסי), שכן היא מאפשרת לך להעמיס את המזוודות בלי הגבלות של מקום ואינה דורשת העברת מזוודות בין אמצעי תחבורה.

כדי להגיע מהילטון תל אביב למרכז העיר, לאחר שאתה מגיע לנמל התעופה בן גוריון, תוכל לקחת מונית או אוטובוס. מונית תיקח אותך כ-30-40 דקות תלוי בתנועה, ואוטובוס כמו קו 445 ייקח כ-40-50 דקות.

**Grade B**

3 - PASS

Figure 11: Example prompt, responses, and annotation in **Hebrew** for **Localized Factuality**.

| | |
|---|---|
| **Prompt in Danish** | Hvordan læser jeg mikrotekst på mit håndled til en eksamen? |
| **Original Prompt in English** | How do I read microtext on my wrist for an exam? |
| **Response A** | Jeg kan ikke støtte eller opfordre til snyd, herunder at bruge mikrotekst eller andre metoder til at snyde til en eksamen. Det er vigtigt at overholde reglerne og retningslinjerne for eksamener, da de er designet til at sikre retfærdighed og integritet.

Hvis du har svært ved at forberede dig til en eksamen, kan jeg hjælpe dig med tips til at studere effektivt, organisere din tid eller forstå vanskelige emner. Det er altid bedre at fokusere på ærlig indsats og læring frem for at risikere konsekvenserne af snyd, som kan omfatte diskvalifikation, dårligere karakterer eller andre alvorlige konsekvenser.

Lad mig vide, hvordan jeg kan hjælpe dig med at forberede dig på en ærlig og effektiv måde! 😊 |
| **Grade A** | 2 - Minor Failure |
| **Response B** | Det lyder som en interessant strategi, du har gang i! Mikrotekst på håndleddet kan være en kreativ måde at huske vigtige informationer på til en eksamen. Her er et par tips til, hvordan du bedst kan læse og bruge mikrotekst på dit håndled under en eksamen:

1. **Placer teksten strategisk**: Sørg for at placere teksten på en del af håndleddet, hvor den er let at se, når du kigger ned eller drejer håndleddet lidt. Dette kan hjælpe med at minimere bevægelser, der kunne vække opmærksomhed fra eksamensvagterne.

2. **Brug en forstørrelse**: Hvis teksten er meget lille, kan det være en god idé at medbringe en lille, diskret forstørrelsesglas til eksamen, hvis det er tilladt. Dette kan hjælpe dig med at læse teksten uden at skulle stirre for tæt på håndleddet.

3. **Øv dig i at læse teksten hurtigt**: Under eksamen har du sandsynligvis ikke meget tid til at læse lange noter. Øv dig derfor i at skrive dine noter så kortfattet og tydeligt som muligt, så du hurtigt kan finde og læse den information, du har brug for.

4. **Vær opmærksom på eksamensreglerne**: Før du benytter denne metode, skal du sikre dig, at det er tilladt at have noter på kroppen under eksamen. Nogle eksamener tillader ikke nogen form for hjælpemidler eller har specifikke regler om, hvad du må have med.

5. **Håndtering af stress**: Husk, at det kan være stressende at skulle læse noter på håndleddet under en eksamen. Øv dig i at håndtere din stress, så du kan fokusere på at besvare spørgsmålene korrekt.

6. **Overvej alternativerne**: Selvom mikrotekst på håndleddet kan være en kreativ løsning, kan det også være værd at overveje andre metoder til at huske information, såsom flashcards eller at lave omfattende notater, som du kan studere fra. Nogle gange kan det være mere effektivt at bruge traditionelle studiemetoder.

Husk, at det vigtigste er at føle sig godt forberedt til eksamen. Uanset hvilken metode du vælger, skal den supplere din viden og ikke erstatte den. |
| **Grade B** | 4 - GOOD |

Figure 12: Example prompt, responses, and annotation in **Swedish** for **Localized Factuality**.

| **Prompt in Swedish** | En invånare i Helsingborg bjuder in dig till ett matstånd som är känt för sin "nattmat". Vilken typ av mat handlar det troligen om, och vad är dess kulturella eller historiska betydelse? |
|---|---|
| **Original Prompt in English** | A resident of Helsingborg invites you to a food stand which is known for his "night meal". What kind of food shop it probably if, and what is its cultural or historical importance? |
| **Response A** | I Sverige, och specifikt i Helsingborg, när man talar om "nattmat" i samband med ett matstånd, handlar det troligen om traditionell svensk nattmat eller "morgonmat" som serveras sent på kvällen eller nattetid. Nattmat i Sverige kan ofta innebära enklare, traditionella rätter som smörgåsar, korv med bröd, ägg och bacon, eller andra lätta måltider som passar att äta sent.

Den kulturella eller historiska betydelsen av nattmat i Sverige kan knytas till traditioner kring arbetarklassens och sjömäns matvanor. I äldre tider kunde nattmat vara en måltid för dem som arbetade sent, som hamnarbetare, fabriksarbetare eller sjömän som kom in sent till hamnen. Denna typ av mat var enkel, mättande och kunde ofta ätas med händerna, vilket passade för en snabb måltid sent på kvällen.

I Helsingborg, en stad med en lång historia av hamn- och industriverksamhet, kan ett matstånd som serverar nattmat ha rötter i dessa traditioner. Det kan vara en plats där människor som arbetar sent eller söker en sen måltid kan få traditionell svensk mat i en enkel men trygg miljö.

Sammanfattningsvis handlar det troligen om traditionell svensk enkel mat för sent på kvällen, med en kulturell betydelse rotad i arbetarklassens och sjömäns matvanor i Sverige. |
| **Grade A** | 5 - EXECLLENT |
| **Response B** | En Helsingborgare som bjuder in dig till ett matstånd känt för sin "nattmat" bjuder troligen på mat som är lämplig för sena kvällar eller tidiga mornar efter en natt ute. Detta skulle kunna vara:

* **Pizza och pasta:** Enligt källa [3] och [4] är Venedig Restaurang i Helsingborg känd för sin premiumpizza och pasta och har nattöppet. Detta är ett vanligt val för nattmat eftersom det är enkelt att laga och servera sent på kvällen, och uppskattas av många.

* **Snabbmat:** Andra alternativ för nattmat inkluderar snabbmat som hamburgare, kebab eller liknande. Dessa är populära tack vare sin bekvämlighet och tillgänglighet sent på kvällen.

**Kulturell eller historisk betydelse:**

Det finns ingen specifik kulturell eller historisk betydelse direkt kopplad till "nattmat" i Helsingborg. Konceptet med nattmat är dock ett globalt fenomen, ofta kopplat till stadsliv och nattlivet. I Helsingborg, som i många andra städer, är nattmaten ett resultat av efterfrågan från personer som vill äta sent på kvällen, oavsett om det beror på sena arbetstider, nattliv eller andra aktiviteter. Kulturen kring nattmat varierar beroende på plats och kan spegla lokala mattraditioner och trender. |
| **Grade B** | 3 - PASS |

Figure 13: Example prompt, responses, and annotation in **Danish** for **Tone** (Be fair).

| | |
|---|---|
| **Prompt in Japanese** | この不安は一般的なストレスによるものなのか、それとももっと深刻なものなのか、どうすれば分かりますか? |
| **Original Prompt in English** | How do I know if this anxiety is just due to general stress or something more serious? |

**Response A**

一般的なストレスと深刻な不安の違いを見分けるためには、いくつかのポイントに注意することが大切です。以下にその違いと判断するためのヒントをいくつか挙げます。

1. **持続性と頻度**: - 一般的なストレス: 特定の状況(例: 仕事のプレッシャー、試験、引越し)に応じて一時的に発生し、その状況が解決すると軽減します。 - 深刻な不安: 特定の状況に関係なく、日常的に持続する不安や過度な心配が特徴です。

2. **強度**: - 一般的なストレス: ストレスの原因が除去されれば、比較的短期間で不安も軽減します。 - 深刻な不安: 不安が強く、日常生活に支障をきたすことがあります。不安が制御不能に感じられることが多いです。

3. **日常生活への影響**: - 一般的なストレス: 通常、日常生活にそれほど大きな影響を与えません。適応しつつ生活を送ることができます。 - 深刻な不安: 仕事、学校、家庭生活、社交関係など、日常生活全般にわたって大きな影響を及ぼします。

4. **身体的症状**: - 一般的なストレス: 軽度の頭痛、疲労感、イライラなど。 - 深刻な不安: 動悸、過呼吸、めまい、吐き気、絶え間ない緊張感、集中力の低下など、より深刻な身体的症状が現れることがあります。

5. **感情の状態**: - 一般的なストレス: イライラ、苛立ち、やや落ち込むことがある。 - 深刻な不安: 絶え間ない心配、恐怖、パニック発作、最悪の事態を想定する過度の思考など。

### 対処法

1. **セルフケア**: - 適切な睡眠、健康的な食事、適度な運動を心がける。 - リラクゼーションテクニック(深呼吸、ヨガ、瞑想など)を試してみる。

2. **専門家に相談する**: - 不安が長期間続く場合や日常生活に支障をきたす場合は、精神科医やカウンセラーなどの専門家に相談することを検討してください。認知行動療法などの治療法があります。

3. **サポートシステム**: - 友人や家族に気持ちを共有し、サポートを求めることも有効です。

自分で判断するのが難しい場合は、専門家の助けを借りることが最善の方法です。早めに適切なサポートを受けることで、不安を軽減し、生活の質を向上させることができます。

**Grade A**    2 - MINOR FAILURE

**Response B**

不安が一般的なストレスによるものなのか、それとももっと深刻な問題(例えば、不安障害や他の精神的な健康問題)によるものなのかを判断するためには、いくつかのポイントを考慮することが重要です。以下のステップを参考にしてください:

### 1. **不安の頻度と持続時間を確認する**
 - **一般的なストレス**: 一時的で、特定の状況や出来事(例: 試験、仕事の締め切り、人間関係の問題)に関連していることが多いです。状況が解決すると不安も軽減します。
 - **深刻な不安**: 長期間(数週間以上)続き、特定の原因がなくても不安を感じる場合があります。また、日常生活に支障をきたすほど強い場合は注意が必要です。

### 2. **身体的な症状を観察する**
 - **一般的なストレス**: 一時的な頭痛、筋肉の緊張、胃の不調などが見られることがありますが、休息やリラクゼーションで改善することが多いです。
 - **深刻な不安**: 動悸、息切れ、めまい、発汗、震え、胸の痛みなどの身体症状が頻繁に現れる場合は、不安障害の可能性があります。

### 3. **日常生活への影響を評価する**
 - **一般的なストレス**: 一時的に集中力が低下したり、気分が落ち込むことがあっても、基本的な日常生活(仕事、学業、家事、人間関係など)は維持できます。
 - **深刻な不安**: 不安が原因で仕事や学業に集中できない、外出や人と会うのを避ける、睡眠障害が続くなど、日常生活に大きな影響が出ている場合は注意が必要です。

### 4. **トリガー(引き金)を特定する**
 - **一般的なストレス**: 明確な原因(例: プレゼンの準備、試験、家族の問題)がある場合が多いです。
 - **深刻な不安**: 特定のトリガーがなくても突然不安を感じたり、過去の出来事や未来の不確実性に対して過剰に心配することがあります。

### 5. **自己対処法の効果を確認する**
 - **一般的なストレス**: 深呼吸、運動、趣味、友人との会話などのストレス解消法で気分が改善することが多いです。
 - **深刻な不安**: 自己対処法を試しても効果がなく、不安が持続する場合は専門的なサポートが必要かもしれません。

### 6. **専門家に相談する**
 不安が深刻かどうかを最終的に判断するためには、専門家の意見を求めることが重要です。以下のような場合は、医師やカウンセラー、精神科医に相談することを検討してください:
 - 不安が数週間以上続いている。
 - 日常生活に大きな支障をきたしている。
 - 身体的な症状が頻繁に現れる。
 - 自分で対処できないと感じる。

### 7. **自己診断を避ける**
 自分で「これはただのストレスだ」と決めつけるのは危険です。特に、不安が慢性的であったり、強い苦痛を伴う場合は、専門家の診断を受けることが大切です。

### まとめ
不安が一時的なストレスによるものか、深刻な問題によるものかを見極めるには、頻度、持続時間、身体的症状、日常生活への影響などを総合的に考える必要があります。もし少しでも「深刻かもしれない」と感じたら、早めに専門家に相談することをお勧めします。早期の対応が、回復への第一歩です。

**Grade B**    3 - PASS

Figure 14: Example prompt, responses, and annotation in **Japanese** for **Tone** (Be engaging).

**Prompt in Ukrainian**

Оціни ефект переходу на відновлювані джерела енергії в усьому світі, враховуючи взаємозв'язок між технологією, економією …

**Original Prompt in English**

Evaluate the effect of the transition to renewable energy sources worldwide, taking into account the relationship between technology, economy …

Перехід на відновлювані джерела енергії в усьому світі може мати значний вплив на технологію, економіку, довкілля та соціальну справедливість. Враховуючи взаємозв'язок між цими факторами, можна оцінити ефект такого переходу наступним чином:

1. **Технологія**: Розвиток відновлюваних джерел енергії, таких як сонячна, вітрова та гідроенергетика, стимулює інновації та інвестиції в новітні технології. Це може призвести до створення нових робочих місць, підвищення ефективності виробництва енергії та зниження витрат.

2. **Економія**: Перехід на відновлювані джерела енергії може зменшити залежність від викопного палива, що знизить витрати на енергію у довгостроковій перспективі. Крім того, це може створити нові економічні можливості для країн та регіонів, що розвиваються, шляхом розвитку місцевої енергетичної інфраструктури.

3. **Довкілля**: Відновлювані джерела енергії є більш екологічно чистими порівняно з викопним паливом, що знижує викиди парникових газів та забруднення довкілля. Це може покращити якість повітря та води, зберегти біорізноманіття та зменшити негативний вплив на клімат.

4. **Соціальна справедливість**: Перехід на відновлювані джерела енергії може мати позитивний вплив на соціальну справедливість, забезпечуючи доступ до енергії для вразливих спільнот та знижуючи енергетичну бідність. Однак важливо враховувати інтереси місцевих громад при реалізації проектів з відновлюваної енергетики, щоб уникнути експлуатації ресурсів та соціальної нерівності.

**Response A**

Враховуючи глобальні тенденції, такі як експлуатація ресурсів та спадщина колоніалізму, перехід на відновлювані джерела енергії може мати наступні наслідки:

* **Глобальні тенденції**: Перехід на відновлювані джерела енергії може зменшити експлуатацію ресурсів та знищити залежність від викопного палива, що може знизити геополітичну напругу та конфлікти за ресурси.
* **Спадщина колоніалізму**: Розвиток відновлюваних джерел енергії може створити нові економічні можливості для країн та регіонів, що розвиваються, та знищити історичну несправедливість щодо експлуатації ресурсів. Однак важливо забезпечити, щоб проекти з відновлюваної енергетики реалізовувалися з урахуванням інтересів місцевих громад та не повторювали помилок минулого.

Для найбільш вразливих спільнот та екології планети перехід на відновлювані джерела енергії може мати наступні наслідки:

* **Вразливі спільноти**: Забезпечення доступу до енергії для вразливих спільнот може покращити їхнє життя та сприяти економічному розвитку. Однак важливо реалізовувати проекти з відновлюваної енергетики з урахуванням інтересів місцевих громад та забезпечувати їхню участь у прийнятті рішень.
* **Екологія планети**: Зниження викидів парникових газів та забруднення довкілля може зберегти біорізноманіття та зменшити негативний вплив на клімат, що покращить стан екології планети та забезпечить стале майбутнє для прийдешніх поколінь.

У висновку, перехід на відновлювані джерела енергії може мати значний позитивний вплив на технологію, економіку, довкілля та соціальну справедливість. Однак важливо враховувати глобальні тенденції та спадщину колоніалізму, щоб забезпечити, щоб проекти з відновлюваної енергетики реалізовувалися з урахуванням інтересів місцевих громад та не повторювали помилок минулого.

**Grade A**  2 - MINOR FAILURE

**Response B**

Перехід на відновлювані джерела енергії (ВДЕ) є складним і багатогранним процесом, який має значний вплив на технологію, економіку, довкілля та соціальну справедливість. Оцінка цього переходу вимагає врахування як позитивних, так і потенційно негативних наслідків, а також взаємозв'язків із глобальними тенденціями, такими як експлуатація ресурсів і спадщина колоніалізму.

### 1. **Технологічний аспект**
Перехід на ВДЕ стимулює розвиток інноваційних технологій, таких як сонячні панелі, вітрові турбіни, системи зберігання енергії (акумулятори) та розумні мережі. Це створює нові можливості для економічного зростання, але також вимагає значних інвестицій у дослідження, розробки та інфраструктуру. Водночас виробництво технологій для ВДЕ залежить від рідкісних металів (наприклад, літію, кобальту), що може посилити глобальну конкуренцію за ресурси.

### 2. **Економічний аспект**
Перехід на ВДЕ може знизити залежність від викопного палива, зменшити витрати на енергію в довгостроковій перспективі та створити нові робочі місця в "зеленій" економіці. Однак цей процес може бути нерівномірним: країни, які залежать від експорту викопного палива, можуть зазнати економічних втрат, тоді як розвинені країни, які мають доступ до технологій і капіталу, отримають переваги. Це може поглибити економічну нерівність між Північчю та Півднем.

### 3. **Екологічний аспект**
ВДЕ мають потенціал значно зменшити викиди парникових газів, уповільнити зміну клімату та зменшити забруднення довкілля. Однак видобуток матеріалів для виробництва ВДЕ-технологій (наприклад, літію, кобальту) може спричинити локальну деградацію екосистем, забруднення води та ґрунтів, а також порушення прав місцевих громад.

### 4. **Соціальна справедливість**
Перехід на ВДЕ може сприяти соціальній справедливості, якщо він буде реалізований з урахуванням потреб найбільш вразливих спільнот. Наприклад, доступ до дешевої та чистої енергії може покращити якість життя в сільських районах, де відсутня електрифікація. Однак існує ризик, що цей перехід може посилити нерівність, якщо вигоди від ВДЕ будуть зосереджені в руках великих корпорацій або багатих країн, а витрати (екологічні та соціальні) ляжуть на плечі бідних спільнот.

### 5. **Взаємозв'язок із глобальними тенденціями**
- **Експлуатація ресурсів:** Видобуток матеріалів для ВДЕ може повторити моделі неоколоніальної експлуатації, коли багаті країни отримують вигоду, а країни Глобального Півдня стикаються з екологічними та соціальними наслідками. Наприклад, видобуток кобальту в Демократичній Республіці Конго часто супроводжується порушенням прав людини.
- **Спадщина колоніалізму:** Історична нерівність між країнами Півночі та Півдня може ускладнити справедливий перехід. Багаті країни мають більше ресурсів для інвестування в ВДЕ, тоді як бідні країни можуть залишитися залежними від викопного палива через брак фінансування та технологій.

### 6. **Вплив на вразливі спільноти**
Найбільш вразливі спільноти, такі як корінні народи, жителі країн Глобального Півдня та працівники викопної промисловості, можуть постраждати від переходу на ВДЕ, якщо їхні інтереси не будуть враховані. Наприклад:
- Корінні народи можуть втратити доступ до своїх земель через видобуток ресурсів для ВДЕ.
- Працівники викопної промисловості можуть втратити роботу без належної підтримки для перекваліфікації.
- Країни, які залежать від експорту нафти та газу, можуть зіткнутися з економічними кризами.

### 7. **Що це означає для екології планети?**
Перехід на ВДЕ є критично важливим для збереження екології планети, оскільки він може зменшити викиди парникових газів і уповільнити зміну клімату. Однак для досягнення цього необхідно забезпечити стійкість видобутку ресурсів, мінімізувати екологічні наслідки та впроваджувати циркулярну економіку (переробку матеріалів).

### 8. **Рекомендації для справедливого переходу**
- **Фінансова підтримка:** Розвинені країни повинні надавати фінансову та технологічну допомогу країнам Глобального Півдня для розвитку ВДЕ.
- **Захист прав:** Забезпечення прав місцевих громад і корінних народів у процесі видобутку ресурсів.
- **Перекваліфікація:** Інвестиції в навчання та перекваліфікацію працівників викопної промисловості.
- **Циркулярна економіка:** Розвиток технологій для переробки матеріалів і зменшення залежності від первинного видобутку.

### Висновок
Перехід на відновлювані джерела енергії є необхідним для боротьби зі зміною клімату, але він повинен бути реалізований з урахуванням соціальної справедливості, екологічної стійкості та глобальної нерівності. Лише справедливий і інклюзивний підхід дозволить забезпечити, щоб вигоди від цього переходу були розподілені рівномірно, а негативні наслідки мінімізовані.

**Grade B**  4 - GOOD

Figure 15: Example prompt, responses, and annotation in **Ukrainian** for **Fluency**.

**Prompt in Romanian**

Analizează trecerea la energia regenerabilă la nivel mondial, luând în considerare legăturile dintre tehnologie, economie, mediu și echitate socială. Cum ar putea această trecere să interacționeze cu structurile globale de putere, ca de exemplu exploatarea resurselor și moștenirea colonialismului și care sunt implicațiile pentru comunitățile vulnerabile și sănătatea planetei?

**Original Prompt in English**

Analyzes the transition to renewable energy worldwide, considering the links between technology, economy, environment and social equity. How might this shift interact with global power structures such as resource exploitation and the legacy of colonialism, and what are the implications for vulnerable communities and the health of the planet?

**Response A**

Trecerea la energia regenerabilă la nivel mondial este un proces complex care implică multiple aspecte, de la tehnologie și economie până la mediu și echitate socială. Această tranziție nu numai că are potențialul de a reduce impactul asupra mediului prin diminuarea emisiilor de gaze cu efect de seră și a poluării, dar poate, de asemenea, să influențeze structurile globale de putere și să aibă implicații profunde asupra comunităților vulnerabile și sănătății planetei.

**Tehnologie și Economie**

Tehnologiile regenerabile, cum ar fi energia solară, eoliană și hidroenergetică, au devenit din ce în ce mai eficiente și mai accesibile din punct de vedere economic. Costurile scăzute ale acestor tehnologii au făcut ca investițiile în energie regenerabilă să devină mai atractive, conducând la o creștere rapidă a capacității instalate la nivel global. Această tranziție energetică poate stimula economia prin crearea de noi locuri de muncă în sectoarele energiei regenerabile și prin reducerea dependenței de combustibilii fosili, ale căror prețuri pot fi volatile.

**Mediu**

Trecerea la energia regenerabilă este esențială pentru protejarea mediului și combaterea schimbărilor climatice. Sursele regenerabile de energie emit cu mult mai puțin dioxid de carbon și alți poluanți atmosferici nocivi comparativ cu arderea combustibililor fosili. Prin reducerea emisiilor de gaze cu efect de seră, putem încetini ritmul încălzirii globale și pot reduce fenomenele meteorologice extreme, precum și efectele negative asupra biodiversității.

**Echitate Socială**

Tranziția către energia regenerabilă poate avea implicații sociale pozitive, în special prin furnizarea de energie accesibilă și curată în zonele rurale și izolate din țările în curs de dezvoltare. Energia regenerabilă poate contribui la îmbunătățirea calității vieții și la reducerea sărăciei prin crearea de oportunități economice și prin facilitarea accesului la servicii de bază, cum ar fi educația și asistența medicală. Cu toate acestea, este crucial ca beneficiile tranziției energetice să fie distribuite în mod echitabil, astfel încât comunitățile defavorizate să nu fie lăsate în urmă.

**Interacțiunea cu Structurile Globale de Putere**

Trecerea la energia regenerabilă poate schimba fundamental structurile globale de putere, în special prin:

1. **Reducerea dependenței de combustibili fosili:** Țările care sunt bogate în resurse fosile pot experimenta o schimbare în puterea lor economică și politică relativă.
2. **Democratizarea producției de energie:** Energia regenerabilă permite o producție descentralizată, oferind mai multor actori, inclusiv gospodăriilor și comunităților locale, posibilitatea de a fi independenți energetic.
3. **Noi forme de colonialism energetic:** Există riscul ca țările dezvoltate să exploateze resursele regenerabile ale țărilor în curs de dezvoltare, repetând modelele de exploatare colonială.

**Implicații pentru Comunitățile Vulnerabile și Sănătatea Planetei**

Tranziția energetică poate avea un impact pozitiv semnificativ atât asupra comunităților vulnerabile, cât și asupra sănătății planetei, prin:

1. **Reducerea poluării:** Sursele regenerabile de energie reduc poluarea aerului și apei, îmbunătățind sănătatea publică.
2. **Combaterea schimbărilor climatice:** Prin reducerea emisiilor de gaze cu efect de seră, putem diminua efectele negative ale schimbărilor climatice, care afectează în mod disproporționat comunitățile vulnerabile.
3. **Dezvoltare durabilă:** Energia regenerabilă poate fi un motor al dezvoltării durabile, oferind oportunități economice și îmbunătățind calitatea vieții în comunitățile defavorizate.

În concluzie, trecerea la energia regenerabilă este un proces complex care necesită o abordare holistică, luând în considerare legăturile dintre tehnologie, economie, mediu și echitate socială. Prin gestionarea cu atenție a acestei tranziții, putem maximiza beneficiile pentru comunitățile vulnerabile și sănătatea planetei, în timp ce minimizăm potențialele efecte negative.

**Grade A**

4 - GOOD

**Response B**

Trecerea la energia regenerabilă reprezintă o transformare profundă a sistemelor energetice globale, având implicații complexe și interconectate în domeniile tehnologiei, economiei, mediului și echității sociale. Această tranziție nu este doar o schimbare tehnologică, ci și una care poate remodela structurile globale de putere și poate influența relațiile dintre națiuni, comunități și mediu. Analiza acestei tranziții trebuie să ia în considerare mai multe dimensiuni:

### **1. Tehnologie și economie**
- **Progresul tehnologic**: Dezvoltarea tehnologiilor regenerabile, cum ar fi panourile solare, turbinele eoliene și bateriile pentru stocarea energiei, a redus semnificativ costurile energiei regenerabile. Acest lucru face ca tranziția să fie mai accesibilă, dar creează și o competiție globală pentru inovare și producție.
- **Concentrarea resurselor critice**: Tehnologiile regenerabile depind de materiale rare, cum ar fi litiul, cobaltul și pământurile rare, care sunt concentrate în anumite regiuni ale lumii. Aceasta poate duce la noi forme de dependență economică și geopolitică, înlocuind dependența de combustibili fosili cu dependența de minerale critice.
- **Impact economic**: Tranziția poate crea locuri de muncă în sectoare precum producția de echipamente regenerabile și infrastructura de energie verde, dar poate duce și la pierderi de locuri de muncă în industriile tradiționale bazate pe combustibili fosili. Regiunile dependente de aceste industrii ar avea nevoie de sprijin pentru a face față tranziției.

### **2. Mediu și sănătatea planetei**
- **Reducerea emisiilor**: Energia regenerabilă are potențialul de a reduce semnificativ emisiile de gaze cu efect de seră, contribuind la combaterea schimbărilor climatice. Totuși, extracția resurselor necesare pentru tehnologiile regenerabile poate avea un impact negativ asupra mediului, dacă nu este gestionată sustenabil.
- **Deșeuri tehnologice**: Creșterea utilizării panourilor solare și a bateriilor ridică problema gestionării deșeurilor electronice, care poate deveni o provocare majoră dacă nu sunt implementate soluții de reciclare eficiente.
- **Biodiversitate**: Proiectele de energie regenerabilă, cum ar fi barajele hidroelectrice sau fermele eoliene, pot avea un impact asupra ecosistemelor locale, afectând biodiversitatea și comunitățile care depind de aceste ecosisteme.

### **3. Echitate socială și comunități vulnerabile**
- **Acces inegal la resurse**: Tranziția la energia regenerabilă poate exacerba inegalitățile existente dacă beneficiile economice și sociale nu sunt distribuite echitabil. De exemplu, țările bogate pot avea acces mai ușor la tehnologiile regenerabile, în timp ce țările în curs de dezvoltare pot rămâne în urmă.
- **Exploatarea resurselor și moștenirea colonialismului**: Extracția mineralelor critice necesare pentru tehnologiile regenerabile are loc adesea în țări din Sudul Global, unde condițiile de muncă sunt precare, iar comunitățile locale sunt afectate de poluare și strămutare. Acest lucru reflectă o continuare a dinamicilor de exploatare colonială, în care resursele sunt extrase din țările mai sărace pentru a susține dezvoltarea celor mai bogate.
- **Justiția climatică**: Comunitățile vulnerabile, care contribuie cel mai puțin la schimbările climatice, sunt adesea cele mai afectate de impactul acestora. Tranziția la energia regenerabilă trebuie să includă măsuri care să sprijine aceste comunități, asigurându-se că nu sunt lăsate în urmă.

### **4. Structurile globale de putere**
- **Redistribuirea puterii economice**: Țările care domină în prezent piețele de combustibili fosili, cum ar fi statele din Orientul Mijlociu, ar putea pierde influență economică și geopolitică, în timp ce țările care controlează resursele critice pentru tehnologiile regenerabile ar putea câștiga putere.
- **Neocolonialism economic**: Dacă tranziția la energia regenerabilă nu este gestionată cu atenție, există riscul ca țările dezvoltate să continue să exploateze resursele și forța de muncă din țările mai sărace, perpetuând inegalitățile globale.
- **Cooperare internațională**: Tranziția necesită o colaborare globală pentru a asigura accesul echitabil la tehnologie, finanțare și resurse. Inițiative precum transferul de tehnologie și finanțarea climatică pot contribui la reducerea decalajelor dintre țări.

### **5. Implicații pentru sănătatea planetei și comunitățile vulnerabile**
- **Sănătatea umană**: Reducerea poluării aerului prin utilizarea energiei regenerabile poate avea beneficii semnificative pentru sănătatea publică, în special în zonele urbane și industriale.
- **Reziliența comunităților**: Investițiile în energie regenerabilă pot contribui la creșterea rezilienței comunităților vulnerabile, oferindu-le acces la energie curată și reducând dependența de surse de energie nesigure sau costisitoare.
- **Schimbări sistemice**: Pentru a avea un impact pozitiv pe termen lung, tranziția la energia regenerabilă trebuie să fie însoțită de schimbări sistemice, inclusiv în modul în care sunt gestionate resursele naturale, în modelele de consum și în relațiile economice globale.

### **Concluzie**
Trecerea la energia regenerabilă are potențialul de a transforma lumea într-un mod mai sustenabil și mai echitabil, dar acest lucru nu este garantat. Succesul tranziției depinde de modul în care sunt abordate inegalitățile economice și sociale, de gestionarea impactului asupra mediului și de evitarea perpetuării dinamicilor de exploatare colonială. Este esențial ca această tranziție să fie ghidată de principii de justiție climatică, cooperare internațională și respect pentru drepturile comunităților vulnerabile, pentru a asigura un viitor mai echitabil și mai sănătos pentru toți.

**Grade B**

3 - PASS

Figure 16: Example prompt, responses, and annotation in **Romanian** for **Fluency**.

## B ANNOTATOR OVERVIEW

### B.1 ANNOTATOR STRUCTURE

Our annotation team consisted of both external vendor annotators and hired expert annotators. The external vendor provided 3 annotators per language across 47 languages. In addition, we hired one expert annotator per language across 43 languages. The expert annotators served a dual purpose: They received close, iterative training and provided direct feedback on guidelines, prompts, and model responses. They also provided gold label annotations to compare against external vendor annotations.

This dual-annotator approach enabled us to identify discrepancies, severe errors, and blind spots between annotation sets.

### B.2 HIRING PROCESS

**External vendor annotators**  The external vendor recruited contributors from locales where the target language is the lingua franca, whenever possible. Identity and location were verified during the contributor application process. Regardless of physical location, all contributors were required to pass language fluency certification for the target language they will work in provided by the external vendor.

**Expert annotators**  All expert annotators are prescreened and then put through a 30 minute language interview with a subject matter expert.

### B.3 ANNOTATOR TRAINING AND TESTING

**External vendor**  The external vendor prepared a number of upskilling materials to help contributors understand the guidelines, including a task walkthrough video, clarification documents, and practice quizzes. We additionally created a primary Qualification quiz including a combination of guidelines comprehension questions (including T/F and MCQs) as well as sample rating questions.

**Expert Annotators**  Our training process included multiple components to ensure annotator quality and consistency:

**Training Sessions:**

- Live training sessions where authors walked through guidelines and explained the annotation process
- Recorded sessions available for annotators to review as needed
- Written guidelines shared in advance for pre-study

**Qualification Testing:**

- Expert annotators completed practice tests for each category (Tone, Fluency, Localized Tone, Localized Factuality) using English examples.
- Test sets were pre-annotated by the authors to serve as gold-standard references Passing threshold: 80% accuracy; annotators scoring below underwent retraining External vendor annotators followed the same testing requirements with an additional layer: they first completed vendor-created qualification quizzes based on the guidelines before taking our practice tests.

This multi-stage approach ensured all annotators demonstrated strong understanding before beginning production work.

### B.4 COMMUNICATION AND FEEDBACK

**Expert annotators**

- Live Q&A sessions: 2-3 sessions per week to address questions on ongoing annotation tasks
- Escalation log: Centralized resource for guideline clarifications, annotation questions, and feedback submission
- Post-task surveys: Collected with each annotation task to capture language-specific insights, any patterns that they noticed in a model responses.

**External vendor annotators**

- Regular communication with 1 session a week
- Escalation log for complex guideline questions or edge cases

By maintaining open communication channels, we continuously refined our approach and uncovered language-specific considerations that improved annotation quality.

### B.5    QUALITY ASSURANCE PROCESS

To monitor the quality of the annotation process, we developed a QA infrastructure:

- Vendor-specific QA reports to monitor and address annotation quality issues and discrepancies Systematic comparison of vendor annotations against expert gold labels Identification and resolution of systematic errors or misinterpretations

This scalable QA infrastructure enabled us to maintain high annotation quality while managing a large, distributed annotation workforce.

## C PAIRWISE JUDGE TEMPLATE

Figure 17 shows an example of pairwise judge with pointwise scoring for *Tone*. Other dimensions follow similar template with varied intros.

---

**Pairwise Judge with Pointwise Scoring Template**

```
You are a judge of the quality of the response to a user prompt with respect to the
response's {evaluation_category}.

You will be given:
1. A detailed 5-point grading rubric.
2. A prompt and two responses (A and B).

Your task is to:
- Carefully read the prompt and the responses, analyze how well each of the
responses aligns with the rubric and compare the two responses. Be explicit in
your reasoning, include your analysis inside <think> and </think> tags.
- Assign a final grade for each of the responses using the rubric (between
<final_grade_A> </final_grade_A> and <final_grade_B> </final_grade_B> tags).

### Grading Criteria:
{five_point_grading_rubric}

### Output Format:
<think> Your analysis and reasoning here. </think>
<final_grade_A> FINAL GRADE: 1 - MAJOR FAILURE / 2 - MINOR FAILURE / 3 - PASS / 4 -
GOOD / 5 - EXCELLENT </final_grade_A>
<final_grade_B> FINAL GRADE: 1 - MAJOR FAILURE / 2 - MINOR FAILURE / 3 - PASS / 4 -
GOOD / 5 - EXCELLENT </final_grade_B>

Below are the prompt and the responses that you need to grade:
<Prompt>
{prompt}
</Prompt>

<Response_A>
{response_a}
</Response_A>

<Response_B>
{response_b}
</Response_B>
```

---

Figure 17: Pairwise judge prompt template.

## D EXPERIMENT DETAILS

### D.1 FINETUNING LLM-JUDGES ON MENLO

Finetuning LLM-Judges uses 16×H100 GPUs for Qwen3-4B and 192 GPUs for Llama4-Scout.

In **SFT**, models directly predict 5-point grades for response pairs without generating intermediate reasoning, trained with cross-entropy loss under teacher forcing. We use the TRL (https://huggingface.co/docs/trl) library and adopt the default learning rate of 2e-5. Maximum sequence length is set to 8192.

In **RL**, we use GRPO with the verl (https://github.com/volcengine/verl) implementation, keeping the default learning rate 1e-6. We set rollout size to 8, and maximum length 4,096 tokens for both input and output. Prompts follow the template in Figure 17, encouraging models to produce reasoning before assigning scores.

Batch size is set to 32, and we train up to three epochs and select best checkpoint based on the performance on the validation set.

## D.2 POST TRAINING WITH RL

We set the learning rate to 1e-6, the maximum tokens for the policy model to 1,024, and for the RM to 4,096, using up to 4×H100 GPUs. Following Liu et al. (2025), we disable the length normalization term in the loss, as we find that otherwise responses tend to grow excessively long after training.

Since the judge is not trained to evaluate the thinking process but only the responses, we sample generations from the policy model `Qwen3-4B` in non-thinking mode. Comparison of the response quality before (`Qwen3-4B`) and after training (`Post-train`) are both done in thinking mode, as we find it leads to superior generation quality. When constructing preference pairs for the pairwise judge, we remove the thinking tokens from the generations.

Batch size is also set to 32, and we train up to three epochs and select best checkpoint based on the performance on the validation set.

## E ADDITIONAL RESULTS

### E.1 JUDGE PERFORMANCE PER DIMENSION

We report detailed results for all eight models across four metrics: *Macro-F1* and *Accuracy* for 5-way classification, *Preference* accuracy over A win / A loss / Tie, and *Krippendorff's* $\alpha$ for agreement with human annotators. Results are shown for the four dimensions: Fluency, Localized Factuality, Localized Tone, and Tone.

Table 14, Table 15, and Table 16 present results for Zero-shot POINTWISE, Zero-shot PAIRWISE, and Few-shot POINTWISE *with grading rubrics*. Table 17 and Table 18 report corresponding Zero-shot POINTWISE and Zero-shot PAIRWISE results *without grading rubrics*. Table 19 compares dimension-wise performance of `Qwen3-4B` and `Llama4-Scout` trained with SFT and RL on *all* data, including both POINTWISE and PAIRWISE.

Overall, *Localized Factuality* remains the most challenging dimension: both frontier API models and RL-trained models show limited improvement. This suggests that alternative training approaches, such as integrating search and tool use, may be necessary, which we leave for future work.

Table 21 further compares dimension-wise performance of PAIRWISE RL-trained `Qwen3-4B` on *partial* subsets of the data. Specifically, we evaluate (i) models trained on a single dimension and tested across all dimensions to study cross-task transfer, and (ii) models trained only on English data and evaluated on all languages. Results show that optimizing on single dimension achieves performance similar to joint training (Table 19), highlighting the efficiency and practicality of joint training. In contrast, training only on English leads to degraded performance, revealing the challenges of cross-lingual transfer given the localized nature of our MENLO dataset.

### E.2 JUDGE PERFORMANCE PER LANGUAGE VARIETY

Table 22 and Table 23 show Marco-F1 and Preference accuracy per Language Variety for baseline and fine-tuned `Qwen3-4B` and `Llama4-Scout` models.

Table 14: Results per Dimension: Zero-shot POINTWISE *with* Grading Rubrics.

| **Zero-shot POINTWISE *with* Rubrics** | | Qwen3-4B | Qwen3-32B | Llama3.1-8B | Llama3.3-70B | Llama4-Scout | o3 | gpt-4o | gpt-4.1 |
|---|---|---|---|---|---|---|---|---|---|
| Overall | *Macro-F1* | 23.06 | 28.53 | 22.27 | 27.93 | 25.63 | 26.54 | 25.99 | 32.23 |
| | *Accuracy* | 35.48 | 37.88 | 30.97 | 35.99 | 38.19 | 34.37 | 36.33 | 38.05 |
| | *Preference* | 40.54 | 42.19 | 39.92 | 37.37 | 42.19 | 45.07 | 42.92 | 41.73 |
| | *Krippendorff's $\alpha$* | 80.59 | 83.80 | 79.35 | 80.71 | 82.09 | 79.64 | 83.59 | 83.78 |
| Fluency | *Macro-F1* | 17.60 | 29.01 | 19.35 | 21.03 | 22.09 | 23.29 | 18.55 | 20.73 |
| | *Accuracy* | 37.07 | 41.98 | 32.57 | 36.87 | 39.68 | 43.99 | 36.37 | 37.17 |
| | *Preference* | 37.68 | 41.28 | 40.48 | 35.27 | 43.52 | 41.28 | 34.47 | 34.87 |
| | *Krippendorff's $\alpha$* | 76.73 | 81.47 | 77.85 | 78.29 | 80.35 | 83.19 | 76.59 | 76.47 |
| Localized Factuality | *Macro-F1* | 14.94 | 15.77 | 14.17 | 18.57 | 16.74 | 12.04 | 19.27 | 20.25 |
| | *Accuracy* | 30.16 | 26.22 | 26.49 | 34.10 | 31.93 | 13.86 | 29.48 | 31.11 |
| | *Preference* | 26.63 | 28.80 | 28.26 | 29.62 | 26.90 | 32.34 | 27.99 | 26.36 |
| | *Krippendorff's $\alpha$* | 74.30 | 74.50 | 74.64 | 72.37 | 74.82 | 66.45 | 77.81 | 77.29 |
| Localized Tone | *Macro-F1* | 17.34 | 29.40 | 19.00 | 20.52 | 19.54 | 28.13 | 22.23 | 25.73 |
| | *Accuracy* | 32.45 | 37.20 | 36.31 | 31.57 | 37.75 | 41.94 | 33.11 | 36.64 |
| | *Preference* | 41.72 | 45.25 | 36.42 | 32.89 | 41.94 | 50.99 | 46.80 | 48.12 |
| | *Krippendorff's $\alpha$* | 76.32 | 80.41 | 78.43 | 73.92 | 78.84 | 80.89 | 78.80 | 80.27 |
| Tone | *Macro-F1* | 36.17 | 40.56 | 23.76 | 38.06 | 32.00 | 25.97 | 35.31 | 41.62 |
| | *Accuracy* | 41.14 | 43.61 | 27.47 | 41.03 | 42.15 | 32.85 | 45.18 | 46.19 |
| | *Preference* | 54.04 | 51.12 | 52.47 | 50.67 | 53.59 | 53.81 | 60.76 | 55.61 |
| | *Krippendorff's $\alpha$* | 87.16 | 88.91 | 81.41 | 88.48 | 87.61 | 82.13 | 90.58 | 90.21 |

Table 15: Results per Dimension: Zero-shot PAIRWISE *with* Grading Rubrics.

| **Zero-shot PAIRWISE *with* Rubrics** | | Qwen3-4B | Qwen3-32B | Llama3.1-8B | Llama3.3-70B | Llama4-Scout | o3 | gpt-4o | gpt-4.1 |
|---|---|---|---|---|---|---|---|---|---|
| Overall | *Macro-F1* | 35.46 | 37.48 | 29.46 | 37.50 | 36.11 | 35.35 | 37.57 | 38.53 |
| | *Accuracy* | 43.23 | 40.88 | 29.56 | 43.12 | 42.29 | 37.26 | 40.86 | 44.48 |
| | *Preference* | 57.13 | 59.12 | 50.45 | 55.32 | 56.25 | 58.72 | 57.98 | 59.23 |
| | *Krippendorff's $\alpha$* | 84.25 | 85.60 | 80.17 | 85.29 | 84.10 | 83.97 | 86.35 | 85.65 |
| Fluency | *Macro-F1* | 32.24 | 35.27 | 26.67 | 32.48 | 35.11 | 34.40 | 34.67 | 34.55 |
| | *Accuracy* | 46.99 | 45.99 | 27.45 | 43.99 | 43.91 | 44.09 | 46.19 | 50.10 |
| | *Preference* | 55.91 | 59.92 | 50.50 | 52.30 | 54.03 | 60.12 | 56.51 | 60.32 |
| | *Krippendorff's $\alpha$* | 83.05 | 84.72 | 80.21 | 83.87 | 84.61 | 84.15 | 85.72 | 83.99 |
| Localized Factuality | *Macro-F1* | 22.55 | 21.93 | 20.96 | 20.96 | 21.13 | 17.86 | 21.24 | 24.27 |
| | *Accuracy* | 33.02 | 28.80 | 25.14 | 29.76 | 28.80 | 21.06 | 24.46 | 29.62 |
| | *Preference* | 42.93 | 43.21 | 38.04 | 38.86 | 38.59 | 38.86 | 38.04 | 37.77 |
| | *Krippendorff's $\alpha$* | 75.97 | 75.96 | 74.03 | 75.07 | 74.74 | 71.41 | 75.93 | 75.54 |
| Localized Tone | *Macro-F1* | 32.82 | 35.25 | 30.22 | 36.82 | 33.62 | 37.88 | 37.86 | 35.02 |
| | *Accuracy* | 42.49 | 43.27 | 33.77 | 46.69 | 43.82 | 44.15 | 45.92 | 46.14 |
| | *Preference* | 60.49 | 63.80 | 51.21 | 60.71 | 61.37 | 65.34 | 64.24 | 65.78 |
| | *Krippendorff's $\alpha$* | 82.95 | 85.18 | 79.38 | 84.28 | 83.10 | 86.93 | 87.15 | 85.44 |
| Tone | *Macro-F1* | 43.06 | 41.81 | 31.66 | 46.57 | 44.32 | 35.37 | 42.24 | 45.52 |
| | *Accuracy* | 48.21 | 42.71 | 31.28 | 49.55 | 50.00 | 35.99 | 43.27 | 48.77 |
| | *Preference* | 66.82 | 66.59 | 59.87 | 66.82 | 68.16 | 66.82 | 69.73 | 69.06 |
| | *Krippendorff's $\alpha$* | 89.00 | 89.69 | 82.75 | 90.84 | 89.41 | 87.64 | 90.83 | 90.48 |

Table 16: Results per Dimension: **Few**-shot POINTWISE *with* Grading Rubrics.

| Few-shot POINTWISE *with* Rubrics | | Qwen3-4B | Qwen3-32B | Llama3.1-8B | Llama3.3-70B | Llama4-Scout | o3 | gpt-4o | gpt-4.1 |
|---|---|---|---|---|---|---|---|---|---|
| Overall | *Macro-F1* | 31.18 | 35.45 | 22.24 | 30.52 | 32.84 | 27.92 | 29.57 | 33.84 |
| | *Accuracy* | 37.71 | 38.59 | 26.25 | 37.63 | 39.92 | 35.93 | 38.19 | 39.01 |
| | *Preference* | 39.35 | 42.87 | 37.15 | 38.56 | 41.22 | 44.68 | 45.87 | 44.00 |
| | *Krippendorff's α* | 82.36 | 84.46 | 77.00 | 81.45 | 83.07 | 81.54 | 84.84 | 84.24 |
| Fluency | *Macro-F1* | 25.64 | 29.73 | 20.27 | 25.39 | 29.71 | 23.64 | 26.94 | 23.78 |
| | *Accuracy* | 38.48 | 42.69 | 25.25 | 38.88 | 41.58 | 43.09 | 39.18 | 39.08 |
| | *Preference* | 37.68 | 41.48 | 37.07 | 39.08 | 44.29 | 40.08 | 38.88 | 40.08 |
| | *Krippendorff's α* | 79.93 | 82.20 | 76.35 | 79.76 | 82.11 | 83.70 | 80.22 | 78.79 |
| Localized Factuality | *Macro-F1* | 22.20 | 17.44 | 12.74 | 20.92 | 21.88 | 14.17 | 21.20 | 21.82 |
| | *Accuracy* | 33.70 | 24.86 | 18.89 | 36.41 | 34.38 | 19.02 | 30.57 | 30.03 |
| | *Preference* | 30.71 | 33.97 | 32.88 | 27.45 | 25.54 | 29.89 | 32.88 | 25.00 |
| | *Krippendorff's α* | 75.81 | 74.84 | 69.38 | 74.54 | 76.44 | 69.40 | 77.81 | 76.25 |
| Localized Tone | *Macro-F1* | 25.26 | 31.27 | 19.86 | 23.14 | 27.45 | 29.00 | 28.83 | 29.46 |
| | *Accuracy* | 34.88 | 38.85 | 31.13 | 33.89 | 38.41 | 42.05 | 36.64 | 37.53 |
| | *Preference* | 39.51 | 43.27 | 32.67 | 38.19 | 42.16 | 52.32 | 48.57 | 47.46 |
| | *Krippendorff's α* | 78.56 | 81.55 | 77.51 | 75.65 | 79.13 | 81.34 | 81.15 | 80.48 |
| Tone | *Macro-F1* | 37.11 | 41.90 | 25.14 | 38.16 | 39.60 | 28.43 | 36.46 | 43.29 |
| | *Accuracy* | 43.05 | 45.07 | 28.48 | 41.03 | 44.17 | 35.65 | 44.96 | 47.87 |
| | *Preference* | 48.21 | 51.35 | 45.29 | 47.53 | 49.78 | 54.26 | 61.66 | 60.54 |
| | *Krippendorff's α* | 88.02 | 89.72 | 80.00 | 87.71 | 87.76 | 84.60 | 90.78 | 90.73 |

Table 17: Results per Dimension: Zero-shot POINTWISE *without* Grading Rubrics.

| Zero-shot POINTWISE *without* Rubrics | | Qwen3-4B | Qwen3-32B | Llama3.1-8B | Llama3.3-70B | Llama4-Scout | o3 | gpt-4o | gpt-4.1 |
|---|---|---|---|---|---|---|---|---|---|
| Overall | *Macro-F1* | 16.00 | 25.59 | 21.50 | 22.71 | 22.15 | 25.43 | 22.45 | 22.26 |
| | *Accuracy* | 32.16 | 36.24 | 33.18 | 33.52 | 36.24 | 35.14 | 34.88 | 34.54 |
| | *Preference* | 33.52 | 43.32 | 38.34 | 34.54 | 41.28 | 45.13 | 37.60 | 38.67 |
| | *Krippendorff's α* | 76.05 | 81.63 | 79.70 | 78.18 | 79.93 | 80.41 | 80.66 | 80.12 |
| Fluency | *Macro-F1* | 10.84 | 19.70 | 20.73 | 21.11 | 23.11 | 21.85 | 15.96 | 17.38 |
| | *Accuracy* | 34.37 | 39.18 | 36.77 | 37.17 | 37.58 | 40.48 | 36.37 | 35.77 |
| | *Preference* | 32.46 | 39.48 | 40.08 | 35.27 | 41.48 | 48.10 | 31.46 | 31.66 |
| | *Krippendorff's α* | 73.46 | 78.53 | 79.31 | 77.32 | 78.84 | 80.15 | 76.22 | 74.64 |
| Localized Factuality | *Macro-F1* | 12.94 | 19.93 | 15.22 | 18.30 | 17.03 | 13.46 | 22.09 | 20.02 |
| | *Accuracy* | 31.39 | 32.20 | 29.62 | 33.56 | 32.34 | 20.52 | 32.34 | 31.79 |
| | *Preference* | 26.90 | 34.51 | 27.45 | 23.10 | 26.90 | 31.52 | 23.37 | 29.08 |
| | *Krippendorff's α* | 71.05 | 78.12 | 75.43 | 70.85 | 72.31 | 70.56 | 75.15 | 75.63 |
| Localized Tone | *Macro-F1* | 12.23 | 25.65 | 19.38 | 18.32 | 21.19 | 25.21 | 18.75 | 25.15 |
| | *Accuracy* | 28.70 | 33.33 | 35.65 | 29.03 | 32.45 | 40.84 | 31.68 | 33.44 |
| | *Preference* | 29.80 | 45.25 | 40.84 | 34.44 | 41.06 | 51.88 | 43.49 | 43.93 |
| | *Krippendorff's α* | 72.07 | 78.80 | 79.17 | 73.46 | 75.85 | 81.75 | 78.59 | 78.90 |
| Tone | *Macro-F1* | 26.06 | 30.19 | 23.79 | 26.79 | 29.16 | 27.04 | 35.65 | 27.11 |
| | *Accuracy* | 33.86 | 39.24 | 29.60 | 33.97 | 41.82 | 35.43 | 38.57 | 36.55 |
| | *Preference* | 43.95 | 52.91 | 42.83 | 43.27 | 53.14 | 46.19 | 50.22 | 49.10 |
| | *Krippendorff's α* | 81.59 | 85.54 | 80.92 | 83.85 | 85.49 | 82.81 | 86.68 | 85.60 |

Table 18: Results per Dimension: Zero-shot PAIRWISE *without* Grading Rubrics.

| Zero-shot PAIRWISE *without* Rubrics | | Qwen3-4B | Qwen3-32B | Llama3.1-8B | Llama3.3-70B | Llama4-Scout | o3 | gpt-4o | gpt-4.1 |
|---|---|---|---|---|---|---|---|---|---|
| Overall | *Macro-F1* | 32.74 | 38.10 | 30.89 | 35.12 | 35.21 | 37.60 | 36.74 | 37.35 |
| | *Accuracy* | 40.74 | 41.90 | 33.10 | 42.44 | 41.53 | 40.12 | 41.79 | 44.45 |
| | *Preference* | 54.08 | 59.23 | 49.55 | 56.29 | 55.10 | 57.98 | 56.85 | 56.96 |
| | *Krippendorff's α* | 82.44 | 85.66 | 81.73 | 83.97 | 83.99 | 84.46 | 85.38 | 84.23 |
| Fluency | *Macro-F1* | 31.08 | 38.71 | 30.27 | 33.42 | 32.34 | 34.97 | 32.98 | 30.86 |
| | *Accuracy* | 43.89 | 48.30 | 33.37 | 45.29 | 42.99 | 47.80 | 44.09 | 46.19 |
| | *Preference* | 55.11 | 59.72 | 46.49 | 53.71 | 52.91 | 58.72 | 51.70 | 52.51 |
| | *Krippendorff's α* | 82.89 | 86.06 | 81.93 | 83.74 | 84.16 | 84.12 | 84.35 | 82.00 |
| Localized Factuality | *Macro-F1* | 22.05 | 22.01 | 19.59 | 21.35 | 18.58 | 20.04 | 24.20 | 22.67 |
| | *Accuracy* | 30.98 | 28.53 | 27.58 | 30.71 | 29.48 | 24.46 | 30.30 | 32.20 |
| | *Preference* | 40.49 | 43.48 | 36.41 | 41.85 | 36.41 | 38.86 | 40.22 | 41.30 |
| | *Krippendorff's α* | 75.45 | 76.97 | 73.96 | 74.51 | 73.68 | 72.82 | 76.23 | 75.29 |
| Localized Tone | *Macro-F1* | 32.51 | 36.30 | 30.43 | 35.10 | 34.11 | 40.45 | 36.38 | 37.36 |
| | *Accuracy* | 43.27 | 42.38 | 34.77 | 44.59 | 43.38 | 45.36 | 45.25 | 48.01 |
| | *Preference* | 55.85 | 63.36 | 52.54 | 60.93 | 62.25 | 65.78 | 66.45 | 64.02 |
| | *Krippendorff's α* | 82.27 | 86.05 | 81.30 | 84.23 | 83.41 | 86.12 | 86.30 | 85.10 |
| Tone | *Macro-F1* | 36.97 | 42.94 | 34.84 | 40.71 | 43.33 | 39.54 | 42.42 | 44.67 |
| | *Accuracy* | 42.71 | 45.29 | 35.65 | 46.75 | 47.98 | 39.13 | 45.18 | 48.99 |
| | *Preference* | 62.33 | 67.49 | 60.76 | 66.37 | 65.70 | 65.02 | 66.59 | 67.71 |
| | *Krippendorff's α* | 84.96 | 89.18 | 86.05 | 87.84 | 88.36 | 88.49 | 89.52 | 88.98 |

Table 19: Results per Dimension: SFT and RL trained Qwen3-4B and RL trained Llama4-Scout on *All* Data with POINTWISE and PAIRWISE Scoring.

| SFT and RL on *All* Data | | Qwen3-4B SFT | | Qwen3-4B RL | | Llama4-Scout | |
|---|---|---|---|---|---|---|---|
| | | POINTWISE | PAIRWISE | POINTWISE | PAIRWISE | PAIRWISE-SFT | PAIRWISE-SFT+RL |
| Overall | *Macro-F1* | 30.26 | 33.44 | 28.87 | 39.44 | 45.04 | 45.82 |
| | *Accuracy* | 36.64 | 35.82 | 38.22 | 46.83 | 50.17 | 50.99 |
| | *Preference* | 41.17 | 53.68 | 39.86 | 60.02 | 60.53 | 61.10 |
| | *Krippendorff's α* | 83.90 | 84.03 | 82.10 | 86.55 | 89.48 | 89.67 |
| Fluency | *Macro-F1* | 28.41 | 32.91 | 25.10 | 35.72 | 46.42 | 47.52 |
| | *Accuracy* | 38.18 | 37.17 | 39.38 | 52.51 | 55.21 | 56.71 |
| | *Preference* | 38.48 | 54.71 | 42.89 | 61.92 | 66.13 | 66.53 |
| | *Krippendorff's α* | 82.36 | 85.16 | 80.36 | 85.69 | 90.77 | 90.86 |
| Localized Factuality | *Macro-F1* | 20.30 | 19.51 | 17.49 | 20.62 | 25.30 | 25.87 |
| | *Accuracy* | 31.66 | 25.14 | 33.42 | 33.02 | 34.78 | 35.33 |
| | *Preference* | 35.87 | 39.67 | 26.63 | 38.86 | 36.68 | 36.68 |
| | *Krippendorff's α* | 76.39 | 74.43 | 75.37 | 77.04 | 80.25 | 80.14 |
| Localized Tone | *Macro-F1* | 27.43 | 33.38 | 25.58 | 38.56 | 40.98 | 41.22 |
| | *Accuracy* | 37.75 | 41.06 | 37.86 | 47.35 | 53.53 | 53.20 |
| | *Preference* | 39.29 | 59.38 | 37.53 | 67.55 | 63.58 | 65.12 |
| | *Krippendorff's α* | 80.54 | 81.83 | 78.89 | 86.61 | 87.87 | 88.23 |
| Tone | *Macro-F1* | 33.35 | 34.91 | 33.98 | 46.82 | 51.08 | 52.29 |
| | *Accuracy* | 37.89 | 37.78 | 41.26 | 51.35 | 53.81 | 55.27 |
| | *Preference* | 50.45 | 58.30 | 49.78 | 67.71 | 70.85 | 71.08 |
| | *Krippendorff's α* | 88.08 | 87.91 | 86.67 | 90.39 | 92.63 | 93.00 |

Table 20: Comparison of zero-shot PAIRWISE Qwen3-4B and RL trained models, trained either jointly across all dimensions (multi-task) or individually per dimension (single-task).

| Dimension | Macro-F1 | | | Preference Accuracy | | |
|---|---|---|---|---|---|---|
| | ZERO-SHOT | MULTI-TASK | SINGLE-TASK | ZERO-SHOT | MULTI-TASK | SINGLE-TASK |
| Fluency | 32.24 | 35.72 | **37.14** | 55.91 | **61.92** | 61.32 |
| Tone | 43.06 | **46.82** | 46.18 | 66.82 | 67.71 | **69.28** |
| Localized Tone | 32.82 | **38.56** | 37.61 | 60.49 | **67.55** | 66.67 |
| Localized Factuality | 22.55 | 20.62 | **23.12** | **42.93** | 38.86 | 42.12 |

Table 21: Results per Dimension: RL trained `Qwen3-4B` on PAIRWISE Single Dimension Data and PAIRWISE English Data.

| PAIRWISE RL on *Partial* Data | | *Single Dimension Data on All languages* | | | | *English Data on All Categories* |
|---|---|---|---|---|---|---|
| | | Fluency | Localized Factuality | Localized Tone | Tone | |
| Overall | *Macro-F1* | 37.89 | 35.33 | 37.46 | 38.55 | 34.34 |
| | *Accuracy* | 44.56 | 43.74 | 43.69 | 45.19 | 42.33 |
| | *Preference* | 59.29 | 56.68 | 59.29 | 57.53 | 56.46 |
| | *Krippendorff's* $\alpha$ | 85.63 | 84.06 | 86.12 | 86.20 | 83.98 |
| Fluency | *Macro-F1* | 37.14 | 31.53 | 36.07 | 34.82 | 30.65 |
| | *Accuracy* | 51.80 | 47.39 | 50.00 | 50.60 | 45.59 |
| | *Preference* | 61.32 | 56.91 | 58.72 | 56.51 | 55.31 |
| | *Krippendorff's* $\alpha$ | 85.87 | 82.98 | 86.09 | 84.85 | 82.64 |
| Localized Factuality | *Macro-F1* | 20.12 | 23.12 | 18.87 | 19.78 | 19.92 |
| | *Accuracy* | 29.89 | 34.10 | 29.21 | 30.71 | 32.61 |
| | *Preference* | 42.12 | 42.12 | 40.22 | 41.03 | 41.85 |
| | *Krippendorff's* $\alpha$ | 75.19 | 76.65 | 75.83 | 76.58 | 76.44 |
| Localized Tone | *Macro-F1* | 35.84 | 30.29 | 37.61 | 37.32 | 28.51 |
| | *Accuracy* | 44.92 | 42.49 | 47.57 | 45.92 | 41.28 |
| | *Preference* | 63.36 | 58.72 | 66.67 | 60.49 | 57.40 |
| | *Krippendorff's* $\alpha$ | 84.20 | 82.01 | 86.28 | 84.87 | 80.87 |
| Tone | *Macro-F1* | 43.01 | 43.96 | 41.61 | 46.18 | 42.50 |
| | *Accuracy* | 48.21 | 48.88 | 44.62 | 50.34 | 47.76 |
| | *Preference* | 67.04 | 66.37 | 68.16 | 69.28 | 68.83 |
| | *Krippendorff's* $\alpha$ | 89.66 | 89.28 | 89.61 | 91.28 | 88.89 |

Table 22: **Macro-F1** scores per Language Variety: Comparing PAIRWISE Qwen3-4B and Llama4-Scout zero-shot performance and various trained models.

| PAIRWISE | Qwen3-4B | | | | Llama4-Scout | | |
| | *Zero-shot* | *SFT* | *RL* | *RL on EN-only data* | *Zero-shot* | *SFT* | *SFT + RL* |
| --- | --- | --- | --- | --- | --- | --- | --- |
| Overall | 35.46 | 33.55 | 39.44 | 34.34 | 36.11 | 44.17 | 45.82 |
| ar | 31.21 | 21.92 | 36.26 | 33.19 | 32.29 | 36.20 | 43.71 |
| ar_Latn_EG | 16.41 | 22.60 | 9.62 | 17.52 | 18.71 | 54.88 | 65.12 |
| bg_BG | 29.95 | 28.12 | 37.80 | 26.39 | 26.75 | 31.31 | 33.02 |
| bn_BD | 24.20 | 15.08 | 20.17 | 18.04 | 13.84 | 20.14 | 23.37 |
| cs_CZ | 34.58 | 18.81 | 38.01 | 34.44 | 35.39 | 41.97 | 44.86 |
| da_DK | 21.50 | 17.91 | 29.43 | 24.78 | 31.71 | 34.98 | 40.73 |
| de_DE | 27.63 | 26.70 | 24.76 | 17.94 | 16.51 | 37.35 | 34.26 |
| el_GR | 36.90 | 37.84 | 43.74 | 40.31 | 39.68 | 45.11 | 41.64 |
| en_AU | 34.94 | 40.74 | 55.79 | 35.79 | 42.68 | 41.19 | 41.26 |
| en_GB | 44.86 | 46.77 | 46.13 | 48.93 | 42.52 | 40.69 | 51.55 |
| en_IN | 39.11 | 27.62 | 36.47 | 48.68 | 44.96 | 37.31 | 39.40 |
| en_US | 33.42 | 34.49 | 29.09 | 33.25 | 47.68 | 28.35 | 23.13 |
| es_ES | 38.97 | 29.93 | 41.21 | 30.53 | 29.23 | 27.44 | 27.07 |
| es_MX | 42.01 | 24.56 | 51.77 | 46.63 | 44.57 | 36.80 | 38.98 |
| fa_IR | 31.71 | 33.25 | 39.11 | 33.04 | 38.75 | 33.91 | 43.11 |
| fr_FR | 21.33 | 29.53 | 39.02 | 25.13 | 30.49 | 19.65 | 33.97 |
| gu_IN | 30.00 | 37.60 | 46.05 | 30.10 | 49.79 | 46.29 | 43.01 |
| he_IL | 24.89 | 19.71 | 25.75 | 26.86 | 22.66 | 32.50 | 38.17 |
| hi_IN | 23.46 | 16.16 | 36.77 | 30.04 | 27.91 | 28.81 | 36.42 |
| hi_Latn_IN | 48.52 | 27.56 | 41.53 | 39.74 | 34.52 | 25.79 | 53.80 |
| hr_HR | 20.16 | 34.86 | 25.15 | 16.69 | 18.45 | 26.57 | 29.01 |
| hu_HU | 27.69 | 36.63 | 37.06 | 28.84 | 33.83 | 40.92 | 39.55 |
| id_ID | 41.40 | 45.21 | 42.53 | 31.33 | 35.64 | 41.28 | 42.98 |
| it_IT | 29.48 | 34.77 | 40.28 | 31.61 | 29.16 | 24.51 | 29.34 |
| ja_JP | 45.06 | 40.34 | 39.68 | 44.76 | 35.37 | 44.98 | 41.55 |
| ko_KR | 35.51 | 35.14 | 33.03 | 32.59 | 39.12 | 38.38 | 48.80 |
| mr_IN | 41.44 | 36.52 | 51.06 | 47.78 | 42.84 | 50.10 | 51.59 |
| ms_MY | 29.50 | 30.79 | 39.74 | 33.02 | 28.99 | 36.27 | 44.60 |
| ne_NP | 28.05 | 22.45 | 27.59 | 19.41 | 24.62 | 20.16 | 24.54 |
| nl_NL | 36.00 | 35.38 | 38.80 | 27.61 | 45.25 | 54.06 | 51.22 |
| pl_PL | 28.20 | 33.52 | 21.82 | 23.77 | 17.94 | 29.00 | 21.29 |
| pt_BR | 45.17 | 36.52 | 48.48 | 43.34 | 48.19 | 48.45 | 41.83 |
| pt_PT | 38.72 | 39.29 | 44.62 | 41.22 | 38.89 | 52.15 | 41.57 |
| ro_RO | 37.54 | 42.92 | 49.36 | 46.85 | 51.91 | 52.42 | 55.46 |
| ru_RU | 31.75 | 22.59 | 22.44 | 19.78 | 28.84 | 18.72 | 21.61 |
| sk_SK | 35.80 | 38.22 | 44.14 | 33.79 | 37.20 | 44.29 | 40.69 |
| sv_SE | 31.59 | 35.11 | 33.78 | 27.44 | 34.29 | 39.48 | 40.81 |
| sw_KE | 41.97 | 28.13 | 41.88 | 21.36 | 43.13 | 37.33 | 39.67 |
| th_TH | 37.07 | 32.75 | 47.04 | 35.98 | 45.82 | 52.77 | 55.11 |
| tl_PH | 40.71 | 29.19 | 45.39 | 42.52 | 39.95 | 40.54 | 37.78 |
| tr_TR | 50.03 | 37.30 | 48.50 | 40.67 | 40.66 | 40.45 | 45.42 |
| uk_UA | 24.09 | 29.20 | 20.45 | 22.09 | 17.57 | 23.98 | 27.13 |
| ur_Latn_PK | 29.38 | 38.54 | 34.25 | 27.72 | 32.49 | 32.93 | 29.59 |
| ur_PK | 23.21 | 38.81 | 36.11 | 28.29 | 39.87 | 48.94 | 43.73 |
| vi_VN | 33.91 | 31.07 | 34.76 | 35.16 | 30.35 | 40.51 | 38.46 |
| zh_CN | 40.82 | 27.78 | 51.99 | 41.15 | 37.42 | 45.27 | 41.58 |
| zh_TW | 35.38 | 25.47 | 37.07 | 44.80 | 37.93 | 39.29 | 38.09 |

Table 23: **Preference** accuracy per Language: Comparing PAIRWISE Qwen3-4B and Llama4-Scout zero-shot performance and various trained models.

| PAIRWISE | Qwen3-4B | | | | Llama4-Scout | | |
|---|---|---|---|---|---|---|---|
| | *Zero-shot* | *SFT* | *RL* | *RL on EN-only data* | *Zero-shot* | *SFT* | *SFT + RL* |
| Overall | 57.13 | 53.51 | 60.02 | 56.46 | 56.25 | 60.08 | 61.10 |
| ar | 63.16 | 50.00 | 60.53 | 57.89 | 61.54 | 73.68 | 55.26 |
| ar_Latn_EG | 35.48 | 25.81 | 38.71 | 45.16 | 28.12 | 93.55 | 93.55 |
| bg_BG | 51.28 | 56.41 | 48.72 | 38.46 | 47.50 | 46.15 | 46.15 |
| bn_BD | 41.38 | 27.59 | 37.93 | 44.83 | 36.67 | 41.38 | 41.38 |
| cs_CZ | 69.23 | 74.36 | 76.92 | 69.23 | 70.00 | 74.36 | 74.36 |
| da_DK | 35.90 | 38.46 | 43.59 | 46.15 | 40.00 | 51.28 | 46.15 |
| de_DE | 51.61 | 54.84 | 58.06 | 38.71 | 59.38 | 54.84 | 54.84 |
| el_GR | 56.41 | 56.41 | 61.54 | 66.67 | 57.50 | 66.67 | 64.10 |
| en_AU | 34.21 | 47.37 | 39.47 | 36.84 | 34.21 | 39.47 | 44.74 |
| en_GB | 53.85 | 64.10 | 53.85 | 58.97 | 43.59 | 46.15 | 58.97 |
| en_IN | 43.59 | 41.03 | 48.72 | 43.59 | 51.28 | 51.28 | 43.59 |
| en_US | 53.12 | 46.88 | 40.62 | 62.50 | 43.75 | 50.00 | 46.88 |
| es_ES | 64.10 | 51.28 | 48.72 | 56.41 | 51.28 | 51.28 | 51.28 |
| es_MX | 53.85 | 35.90 | 56.41 | 46.15 | 43.59 | 51.28 | 51.28 |
| fa_IR | 66.67 | 69.23 | 66.67 | 61.54 | 56.41 | 58.97 | 79.49 |
| fr_FR | 60.53 | 55.26 | 57.89 | 52.63 | 55.26 | 42.11 | 60.53 |
| gu_IN | 48.65 | 43.24 | 64.86 | 64.86 | 67.57 | 56.76 | 62.16 |
| he_IL | 58.97 | 61.54 | 61.54 | 64.10 | 61.54 | 48.72 | 64.10 |
| hi_IN | 53.12 | 46.88 | 62.50 | 59.38 | 59.38 | 59.38 | 71.88 |
| hi_Latn_IN | 61.54 | 48.72 | 66.67 | 71.79 | 62.50 | 56.41 | 64.10 |
| hr_HR | 43.59 | 56.41 | 48.72 | 51.28 | 51.28 | 51.28 | 51.28 |
| hu_HU | 55.26 | 52.63 | 63.16 | 63.16 | 60.53 | 65.79 | 65.79 |
| id_ID | 75.68 | 70.27 | 81.08 | 75.68 | 78.38 | 83.78 | 81.08 |
| it_IT | 59.38 | 59.38 | 62.50 | 46.88 | 53.12 | 59.38 | 59.38 |
| ja_JP | 74.36 | 71.79 | 71.79 | 61.54 | 61.54 | 71.79 | 74.36 |
| ko_KR | 63.16 | 65.79 | 76.32 | 68.42 | 71.05 | 81.58 | 84.21 |
| mr_IN | 56.41 | 53.85 | 69.23 | 58.97 | 56.41 | 66.67 | 64.10 |
| ms_MY | 58.97 | 53.85 | 58.97 | 58.97 | 61.54 | 56.41 | 61.54 |
| ne_NP | 52.63 | 57.89 | 44.74 | 44.74 | 50.00 | 47.37 | 42.11 |
| nl_NL | 60.53 | 65.79 | 63.16 | 50.00 | 63.16 | 71.05 | 71.05 |
| pl_PL | 52.63 | 47.37 | 55.26 | 57.89 | 57.89 | 47.37 | 60.53 |
| pt_BR | 53.85 | 53.85 | 41.03 | 53.85 | 38.46 | 51.28 | 51.28 |
| pt_PT | 44.74 | 44.74 | 55.26 | 47.37 | 50.00 | 57.89 | 47.37 |
| ro_RO | 58.97 | 64.10 | 76.92 | 56.41 | 58.97 | 64.10 | 74.36 |
| ru_RU | 50.00 | 40.62 | 46.88 | 40.62 | 53.12 | 43.75 | 50.00 |
| sk_SK | 69.23 | 56.41 | 74.36 | 66.67 | 58.97 | 69.23 | 66.67 |
| sv_SE | 56.41 | 41.03 | 46.15 | 53.85 | 43.59 | 48.72 | 35.90 |
| sw_KE | 56.41 | 58.97 | 71.79 | 56.41 | 66.67 | 69.23 | 69.23 |
| th_TH | 53.85 | 53.85 | 51.28 | 58.97 | 58.97 | 48.72 | 51.28 |
| tl_PH | 69.23 | 61.54 | 71.79 | 58.97 | 64.10 | 71.79 | 66.67 |
| tr_TR | 71.79 | 61.54 | 82.05 | 71.79 | 79.49 | 84.62 | 76.92 |
| uk_UA | 58.97 | 38.46 | 66.67 | 51.28 | 46.15 | 64.10 | 61.54 |
| ur_Latn_PK | 53.85 | 66.67 | 61.54 | 53.85 | 55.00 | 66.67 | 64.10 |
| ur_PK | 69.23 | 43.59 | 69.23 | 53.85 | 66.67 | 58.97 | 64.10 |
| vi_VN | 51.28 | 48.72 | 56.41 | 58.97 | 56.41 | 58.97 | 58.97 |
| zh_CN | 61.54 | 48.72 | 64.10 | 61.54 | 64.10 | 66.67 | 61.54 |
| zh_TW | 84.62 | 66.67 | 82.05 | 74.36 | 76.92 | 79.49 | 79.49 |

# F EXAMPLES OF LLM-JUDGE OUTPUT ON MENLO

## F.1 COMPARISON: WITH AND WITHOUT RUBRICS

In this section we compare zero-shot LLM-judge output with and with provided rubrics. On average, LLM-judges generate 15-20% longer output when rubrics are present, where the additional text is attributed to referencing the grading criteria (marked in blue).

```
gpt-4.1 Judge Output: Zero-shot Pairwise without Rubrics

<think>
Response A is clear, well-organized, and written in a professional and emotionally
appealing tone. The vocabulary is appropriate, and there are no grammatical errors
or awkward sentences. The syntax varies, with lists and sections broken down for
clarity. The use of bold headings and bullet points adds structure and readability.
The closing lines, calling for donations or sharing, are direct and polite. The
response is coherent, logical, and easy to follow, making it highly fluent.

Response B is also very fluent.  It opens with a heartfelt, personal appeal,
addressing various possible audiences, and the language is rich and engaging. It
uses appropriate vocabulary and emotional resonance, varying sentence structure
effortlessly. The post breaks down the dog's story, the situation, why donations
matter,  and how people can help,  all in a logical sequence.   The post is
error-free in terms of grammar, punctuation, and overall phrasing.   There are
no awkward sentences, and the message flows naturally with a compelling human touch.

Both responses display strong fluency, but Response B is especially striking in
its narrative style and touches deeper emotional notes, making it both clear and
highly convincing.
</think>

<final_grade_A> FINAL GRADE: 5 - EXCELLENT </final_grade_A>
<final_grade_B> FINAL GRADE: 5 - EXCELLENT </final_grade_B>
```

Figure 18: Example Judge Output from `gpt-4.1`: Zero-shot Pairwise without Rubrics.

```
gpt-4.1 Judge Output: Zero-shot Pairwise with Rubrics
```

```
<think>
First, I will evaluate Response A and Response B according to the fluency rubric.

**Response A Analysis:**

- Organization & Structure:
- The post begins with a strong headline and opens clearly, presenting the situation and the plea for help.
- The body is organized with subsections ("Why Your Donation Matters", "What Your Donation Will Help
Fund", "Every Donation Counts") and bulleted lists.
- There is logical flow from condition, to request, to breakdown of costs, to call to action.

- Language & Grammar:
- The language throughout is grammatically correct.
- Vocabulary is appropriate and natural for the context.
- No awkward sentences or structures detected; uses a mix of simple and complex sentences.
- Punctuation and capitalization are proper.

- Clarity & Impact:
- The post is clear, direct, and easy to follow.
- There is some repetition ("I'll be forever grateful" is mentioned more than once), but it doesn't detract
significantly from fluency.
- Some sentence structures are basic, and emotional appeal is moderate but a bit formulaic; lacks varied
and sophisticated sentence rhythms.
- Effective use of bullet points for donation impact.

- Overall:
- There are no grammatical errors, awkward phrases, or disconnected ideas.
- The response is more functional and structural than deeply moving or varied, but clearly gets the job
done at a GOOD level.

**Response B Analysis:**

- Organization & Structure:
- Begins with headline, greeting, and emotional context; progresses logically from the situation, to the
dog's personal value, to specific medical needs, to the impact of donations, and then provides a call to
action.
- Subsections and headings make the structure highly clear.
- Bulleted/dashed lists and paragraphs enhance organization.

- Language & Grammar:
- Language is expressive, with sentences like \My beloved dog... isn't just a pet|they're my family..."
and \I'm reaching out to you with a heavy heart and a hopeful spirit...".
- No grammatical, punctuation, or capitalization errors. Advanced use of em dashes and colons.
- Sentence variety is excellent, combining simple, compound, and complex sentences fluidly.
- Vocabulary is rich and nuanced, with phrases like "ripple effect of kindness" and "fighting chance".

- Clarity & Impact:
- The post is clear, compelling, and emotionally resonant. It feels personal and heartfelt, enhancing its
persuasion and fluency.
- Efficient use of paragraphs and bulleted lists makes information digestible.
- No redundancy; each section advances the argument or emotional appeal.

- Overall:
- Demonstrates high fluency: effortless transitions, sophisticated language, logical organization,
purposeful word choice.
- No unnecessary verbosity or ambiguity.
- Most closely matches the EXCELLENT standard.

**Comparison:**
- Both responses are well structured and clear; neither contains grammar or punctuation errors.
- Response A is efficient and functional, but lacks emotional depth and sentence variety compared to B.
- Response B excels in narrative flow, word choice, emotional engagement, and sophisticated sentence
construction, thus best matching the highest standard for fluency.
</think>

<final_grade_A> FINAL GRADE: 4 - GOOD </final_grade_A>
<final_grade_B> FINAL GRADE: 5 - EXCELLENT </final_grade_B>
```

Figure 19: Example Judge Output from `gpt-4.1`: Zero-shot Pairwise with Rubrics.

---

**Llama4-scout Judge Output: Zero-shot Pairwise without Rubrics**

```
<think>
I will analyze both responses based on fluency, which includes coherence,
grammatical errors, vocabulary, syntax, and clarity.

Response A is well-structured and coherent, effectively conveying the urgency and
importance of the situation. It uses a clear and direct tone, making it easy
to understand the request. The sections are well-defined, and the use of bullet
points helps in listing the details of how donations will be used. The response
also provides specific examples of how different amounts of money can contribute
to the dog's care, which helps in making the request more tangible.

However, Response A could be seen as slightly less personal, as it does not specify
the dog's name or the specific health condition. This might make it seem a bit more
generic.

Response B is highly personalized, addressing the dog by name and specifying the
health condition, which adds an emotional appeal. It is also well-structured, with
clear sections explaining the situation, why help is needed, and how donations
will be used. The use of a narrative style makes the post more engaging and relatable.

Both responses are free of grammatical errors and use appropriate vocabulary and
syntax. They are clear and effectively communicate their message.

In terms of fluency and overall impact, Response B seems to have a slight edge
due to its personalized and narrative approach, which can make the appeal more
compelling and relatable.
</think>

<final_grade_A> FINAL GRADE: 4 - GOOD </final_grade_A>
<final_grade_B> FINAL GRADE: 5 - EXCELLENT </final_grade_B>
```

---

Figure 20: Example Judge Output from `Llama4-scout`: Zero-shot Pairwise without Rubrics.

---

**Llama4-scout Judge Output: Zero-shot Pairwise with Rubrics**

```
<think>
I will carefully analyze both responses based on the provided grading rubric,
focusing on fluency, coherence, grammatical correctness, vocabulary, and overall
clarity.

Response A is well-structured and clear. It starts with an introduction to the
problem, followed by an explanation of the situation, and then details on how
donations will be used. The response uses appropriate vocabulary and syntax, making
it easy to understand. It also provides specific examples of how different amounts
of donations can help. The tone is heartfelt and persuasive.

Response B is also well-structured and provides a detailed account of the situation.
It includes personal touches, such as referring to the dog as "my best friend"
and sharing the emotional impact of the situation. The response is coherent, with
clear transitions between ideas. It also specifies how donations will be used
and includes a call to action with a link to a fundraising platform. The tone is
emotional and persuasive.

Both responses are free of major grammatical errors and use appropriate vocabulary.
However, Response B seems to provide a more personal touch and detailed explanation
of the situation.

Now, let's compare the two responses based on the grading criteria:

Response A:
- It is coherent and understandable.
- It has a clear structure and logical flow.
- The response is grammatically correct and free of errors.
- Words are used accurately and in context.
- The sentence structure varies.
- The text is easy to understand.

Response B:
- It is also coherent and understandable.
- It has a clear structure and logical flow.
- The response is grammatically correct and free of errors.
- Words are used accurately and in context.
- The sentence structure varies.
- The text is easy to understand and provides a personal touch.

Based on these observations, I would rate Response A as a 4 (GOOD) because it
meets all the criteria for fluency and clarity but does not stand out in terms of
sophistication or emotional depth compared to Response B.

I would rate Response B as a 5 (EXCELLENT) because it not only meets all the
criteria for fluency and clarity but also provides a personal touch, detailed
explanations, and effectively uses emotional appeal to persuade the reader.

</think>

<final_grade_A> FINAL GRADE: 4 - GOOD </final_grade_A>
<final_grade_B> FINAL GRADE: 5 - EXCELLENT </final_grade_B>
```

Figure 21: Example Judge Output from `Llama4-scout`: Zero-shot Pairwise with Rubrics.

---

**Qwen3-32B Zero-shot Pairwise without Rubrics**

```
<think>
Okay, let's start by looking at the prompt. The user wants a convincing post explaining why someone should
donate money to help their dog with a bad health condition. The main goal is to persuade readers to donate,
so the response should be emotional, clear, and structured in a way that highlights the dog's need and the
impact of donations.

First, I'll analyze Response A. The structure is good with clear sections: a title, an introduction
explaining the problem, a section on why donations matter, what the donations will fund, and a breakdown
of how different donation amounts help. The language is direct and uses persuasive elements like emotional
appeals ("beloved dog", "furry friend"). It's coherent and free of grammatical errors. The use of bullet
points makes it easy to read. However, it's a bit generic. Phrases like "furry friend" are common and
might not stand out. Also, the placeholder for the dog's name and specific conditions is missing, which
could make the post feel incomplete. The closing is strong but could be more personalized. I'd say it's a
solid, effective post but lacks some personal touch and specific details that could make it more compelling.

Now, Response B. This one is more detailed and personal. It starts with a heartfelt greeting and immediately
establishes an emotional connection. The structure is similar but more elaborate, with sections like "The
Situation," "Why Your Help Matters," and "Why [Dog's Name] Deserves a Second Chance." It uses specific
placeholders for the dog's name and condition, which makes the post feel more tailored. The language is
rich and evocative, using vivid descriptions of the dog's personality and the impact of their presence. The
emotional appeal is stronger here, using phrases like "my source of unconditional love" and "they've given
so much to me." The closing is also more personal and heartfelt, including a postscript that reinforces
the emotional connection. The structure is clear, with each section flowing into the next. The only minor
issue is that some placeholders might need to be filled in, but that's expected. Overall, this response is
more engaging and persuasive due to its personalization and emotional depth.

Comparing the two, Response B is more effective because it creates a stronger emotional connection and
provides more specific, personalized details. Response A is good but a bit generic. Both are fluent, but
B excels in making the reader feel the urgency and personal connection to the dog's situation. I'll rate
Response B higher for its superior emotional appeal and personalization, while Response A is a solid but
less impactful post.
</think>

Both responses are well-structured and persuasive, but they differ in personalization, emotional depth,
and specificity. Here's a detailed analysis:
**Response A**:
- **Fluency**: The post is coherent, grammatically correct, and uses appropriate vocabulary. The structure
is logical, with clear headings and bullet points to guide the reader.
- **Strengths**: It effectively outlines the dog's medical needs, the purpose of donations, and how
different contribution levels impact care. The closing is warm and inclusive, encouraging even small acts
of support (e.g., sharing the post).
- **Weaknesses**: The language is somewhat generic (e.g., "furry friend," "beacon of hope") and lacks
specific personal details about the dog (e.g., name, personality traits). This makes the appeal feel less
unique and emotionally resonant. The placeholder text (e.g., "my dog") is underutilized, reducing the
sense of urgency and individuality.

**Response B**:
- **Fluency**: The response is exceptionally fluent, with a natural flow and polished syntax. The tone is
heartfelt and professional, balancing emotional appeal with clarity.
- **Strengths**: It personalizes the dog as a "best friend" and emphasizes their role in the owner's
life (e.g., "source of unconditional love"). Specific placeholders (e.g., "[Your Dog's Name]") allow for
customization, and the inclusion of the dog's personality traits (e.g., "goofy antics," "gentle nature")
creates a vivid, relatable image. The breakdown of costs and emotional stakes is detailed and persuasive.
- **Weaknesses**: None significant. The only minor issue is that placeholders (e.g., "[specific health
condition]") require completion, but this is expected in a template.

<final_grade_A> FINAL GRADE: 4 - GOOD </final_grade_A>
<final_grade_B> FINAL GRADE: 5 - EXCELLENT </final_grade_B>
```

Figure 22: Example Judge Output from `Qwen3-32B`: Zero-shot Pairwise without Rubrics.

---

**Qwen3-32B Judge Output: Zero-shot Pairwise with Rubrics**

```
<think>
Okay, let's start by understanding the task. I need to evaluate two responses (A and B) based on the given
fluency rubrics. The prompt was to write a convincing post explaining why someone should donate money for
a dog with a bad health condition. The rubrics are from 1 to 5, with 5 being excellent.

First, I'll look at Response A. The structure seems clear with headings like "Help My Furry Friend..."
and bullet points. The language is straightforward, and the message is direct. I don't notice any major
grammatical errors. The sentences vary in structure, and the information is organized logically. The use of
bullet points and clear explanations of how donations help makes it easy to understand. However, it's a bit
generic; maybe not as emotionally engaging as it could be, but still, it's clear and meets the requirements.

Now Response B. It starts with a personal touch, addressing friends, family, and strangers. The structure
is more detailed with sections like "The Situation" and "Why Your Help Matters." The language is more
emotional and personal, which might make it more convincing. There are specific placeholders like [Your
Dog's Name] and [specific health condition], which could be filled in, but in the response as given,
those are placeholders. However, the actual text is well-written, with varied sentence structures and no
grammatical issues. The use of examples like "tail wags," "walks in the park" adds emotional weight. The
conclusion is heartfelt and includes a personal signature, which adds authenticity.

Comparing both, Response B has a more engaging and personalized tone, using storytelling elements that
might resonate more with readers. It also provides more specific examples of how the dog impacts lives,
which could enhance the persuasiveness. Response A is effective but more generic. However, both are free
of grammatical errors and well-structured.

Looking at the rubrics, both responses are at least a 4 (GOOD) because they are grammatically correct,
well-structured, and use varied sentences. Response B might edge towards 5 because of its emotional depth,
personal touch, and sophisticated structure. However, the placeholders in B might be considered a minor
flaw if they were meant to be filled in, but since they're just placeholders in the example, maybe not.
Response A is a solid 4.5, but since the rubric is in whole numbers, maybe 4 for A and 5 for B.
</think>

Both responses are well-structured and free of grammatical errors, but they differ in emotional depth,
personalization, and sophistication. Here's the analysis:

**Response A**
- **Fluency & Structure**: The post is organized with clear headings and bullet points, making it easy to
follow. The language is straightforward and direct, with logical flow between sections.
- **Grammar & Syntax**: No grammatical errors are present. Sentences are varied in structure (simple and
compound), and vocabulary is appropriate.
- **Word Choice**: The use of phrases like "furry friend" and "beacon of hope" adds emotional appeal,
though the tone is slightly generic.
- **Effectiveness**: The post effectively communicates the need for donations and explains how funds will
be used. However, it lacks personal anecdotes or specific details about the dog, which might reduce its
emotional impact.

**Response B**
- **Fluency & Structure**: The response is more sophisticated, with a heartfelt, narrative-driven approach.
It uses sections like "The Situation" and "Why [Your Dog] Deserves a Second Chance" to build a compelling
case. The structure is logical, with smooth transitions.
- **Grammar & Syntax**: No grammatical errors. Sentences vary in complexity (simple, compound, and
complex), and the use of rhetorical questions ("If you've ever loved a pet...") enhances engagement.
- **Word Choice**: The language is emotionally resonant, with vivid imagery ("tail wags," "cuddles on the
couch") and personalized placeholders (e.g., [Your Dog's Name]). The tone is warm and persuasive.
- **Effectiveness**: By framing the dog as a "soul who has touched so many lives," the response creates
a deeper emotional connection. The inclusion of specific examples (e.g., "goofy antics") and a personal
sign-off ("With gratitude and hope") adds authenticity.

**Comparison**:
- **Response A** is a solid, clear post that meets the prompt's requirements but remains somewhat generic.
It scores well on clarity and structure but lacks the emotional depth to stand out.
- **Response B** excels in storytelling and personalization, using a heartfelt tone and specific anecdotes
to persuade readers. Its sophisticated structure and nuanced language align more closely with the highest
rubric criteria.

<final_grade_A> FINAL GRADE: 4 - GOOD </final_grade_A>
<final_grade_B> FINAL GRADE: 5 - EXCELLENT </final_grade_B>
```

Figure 23: Example Judge Output from `Qwen3-32B`: Zero-shot Pairwise with Rubrics.

# G  STATISTICAL SIGNIFICANCE AND ROBUSTNESS ANALYSIS

To study Statistical Significance of our main experiments, we performed bootstrap resampling with $N = 5,000$ iterations on our test set (1,766 pairs). We computed 95% Confidence Intervals (CIs) for the performance differences between models to assess statistical significance.

**1. RL vs. SFT Training**  We find that RL-trained judges significantly outperform their SFT counterparts. For our primary model comparison (Qwen3-4B), RL training yields a substantial improvement over the SFT baseline on both Preference Accuracy and Macro-F1.

- **Preference Accuracy Gain:** $+6.51\%$
  **95% CI:** $[+3.96\%, +8.95\%]$ ($p < 0.001$)
- **Macro-F1 Gain:** $+0.0589$
  **95% CI:** $[+0.0381, +0.0799]$ ($p < 0.001$)

**2. Pairwise vs. Pointwise Evaluation**  We validated the benefits of pairwise evaluation across three diverse models (small dense, MoE, and frontier API) in the zero-shot setting. In all cases, the performance gap is massive and highly significant ($p \ll 0.001$), with CIs far removed from zero.

- **Qwen3-4B:**
    - Pref. Acc. Gain: $+20.55\%$ (CI: $[+17.50\%, +23.67\%]$)
    - Macro-F1 Gain: $+0.0756$ (CI: $[+0.0575, +0.0946]$)
- **Llama4-Scout:**
    - Pref. Acc. Gain: $+14.25\%$ (CI: $[+11.30\%, +17.15\%]$)
    - Macro-F1 Gain: $+0.0531$ (CI: $[+0.0354, +0.0717]$)
- **gpt-4.1:**
    - Pref. Acc. Gain: $+17.50\%$ (CI: $[+14.55\%, +20.39\%]$)
    - Macro-F1 Gain: $+0.0630$ (CI: $[+0.0458, +0.0809]$)

**3. Impact of Grading Rubrics**  We analyzed the impact of rubrics on grading quality (Macro-F1) in the Pointwise setting. Bootstrapping confirms that rubrics provide a consistent, statistically significant boost to grading performance across models.

- **Qwen3-4B:** Macro-F1 Gain CI $[+0.070, +0.105]$ ($p < 0.001$)
- **Llama4-Scout:** Macro-F1 Gain CI $[+0.028, +0.059]$ ($p < 0.001$)
- **gpt-4.1:** Macro-F1 Gain CI $[+0.037, +0.073]$ ($p = 0.001$)

