# OpenReview forum: "MENLO: From Preferences to Proficiency – Evaluating and Modeling Native-like Quality Across 47 Languages"
_ICLR.cc/2026/Conference — ICLR 2026 Poster_

### Official Review · Reviewer_kNrK · 2025-11-01

**Soundness:** 2
**Presentation:** 3
**Contribution:** 3
**Rating:** 6
**Confidence:** 4

**Summary:**

This paper proposes a framework for evaluating and improving whether large language models generate native-level natural responses across 47 languages. The goal is to evaluate whether models can converse like actual native speakers, beyond mere grammatical accuracy. To this end, the researchers applied the sociolinguistic principle of audience design, which is the concept that speakers adjust their language style according to their addressee or reference group. The framework divides native-level quality into four core dimensions. First, Fluency evaluates accuracy of vocabulary and grammar, consistency of sentence structure, and clarity. Second, Tone judges the overall writing style—whether it is helpful, insightful, engaging, and fair. Third, Localized Tone evaluates alignment with cultural, regional, and linguistic nuances, assessing appropriate use of local expressions and cultural sensitivity. Fourth, Localized Factuality evaluates factual accuracy and completeness, and whether information is provided based on local context. The researchers experimented with whether trained judge models could be used as generative reward models to directly improve the language capabilities of policy models. Using Qwen3-4B-RL-Judge as a reward model to subsequently train the base Qwen3-4B model, all three LLM evaluators (Qwen3-32B, gpt-4.1, Llama4-Scout-RL-Judge) confirmed improvements. Win rates ranged from 63.4% to 77.9%, with average score improvements from +0.80 to +1.16.However, there was an important finding. Human evaluation performed on 10 languages revealed that human evaluators were more conservative with a win rate of 55.7% and average score improvement of +10.9%.

**Strengths:**

Methodological Rigor: Operationalizing sociolinguistic theory (Bell's Style Axiom and audience design) into a computational evaluation framework is a strong point. This goes beyond simply translated prompts or general naturalness evaluation to capture the actual phenomenon of native speakers adjusting language to specific addressees or reference groups. The decomposition into four dimensions is theoretically motivated, and developing detailed rubrics for each dimension effectively reduced subjectivity.

Data Quality and Scale: The MENLO dataset surpasses existing multilingual benchmarks in several aspects. The scale covering 47 language varieties, human-written and localized prompts, and Krippendorff's α of 0.84 are all impressive. Particularly, the fine-grained distinction of language varieties (e.g., European vs. Mexican Spanish, various English varieties) reflects actual linguistic diversity. Deploying expert annotators per language and developing customized annotation tools are also best practices in data quality management.

Systematic Experimentation: Systematic comparisons of pairwise vs. pointwise evaluation, with/without rubrics, and SFT vs. RL clearly show the contribution of each design choice. Particularly, the finding that pairwise evaluation achieves up to 12.4% higher Macro-F1 than pointwise evaluation and is more effective than few-shot examples is significant. Ablation studies on reward shaping also quantify each reward component's contribution, demonstrating the importance of composite reward design in GRPO.

Practical Value: Achieving actual model improvement by using trained judges as reward models demonstrates the framework's practicality. They closed the evaluation-improvement loop, suggesting real applicability beyond theoretical benchmarks. The commitment to releasing datasets and code will also contribute to reproducibility and facilitate follow-up research.

Transparent Limitation Reporting: Honestly reporting the finding that LLM evaluators overestimate improvements by +0.5~0.6 compared to humans demonstrates academic honesty. Acknowledging the difficulty of the Localized Factuality dimension and suggesting alternative approaches (retrieval, tool use) is also positive.

**Weaknesses:**

Insufficient Causal Verification: There's inadequate causal analysis of why pairwise evaluation is superior. It's not distinguished whether comparison itself is important, whether it's due to context richness from simultaneous exposure to two responses, or whether relative judgment is simply easier than absolute judgment. Without control experiments like pairwise evaluation of a single response (presenting the same response twice), mechanistic understanding is limited.

Lack of Statistical Rigor: Most results lack confidence intervals or significance tests. Especially given the high variability in language-specific performance (37.9%~82.1%), it's difficult to judge whether observed differences are statistically significant. Multiple comparisons across several models and settings were performed, so multiple testing corrections like Bonferroni are needed but not applied. Additionally, an average of 137 prompts per language may lack statistical power for some languages.

Missing Comparative Experiments: There's no comparison with existing multilingual quality metrics (BLEURT, COMET, BERTScore), making it difficult to judge MENLO's relative advantages. Also, there's no comparison with other RL algorithms (PPO, DPO, RLAIF) or reward designs, so we cannot know if the proposed GRPO+composite reward is optimal. Human evaluation was only performed on 10 languages, limiting generalization to all 47 languages.

Methodological Bias: The SFT baseline may be unfairly weak. Training without Chain-of-Thought doesn't provide fair comparison with RL, and SFT with reasoning should also be included. Additionally, justification for model selection (Qwen3-4B, Llama4-Scout) is lacking, and result generalization to other sizes or architectures is uncertain.

**Questions:**

I want to express sincere appreciation for the authors' substantial effort in conducting this large-scale, ambitious research. Please consider the following recommendations to strengthen the paper:

< Questions >
Can you provide a more precise operational definition of 'native-level quality'? Please provide theoretical rationale for whether the four dimensions are sufficient or necessary conditions, and why other important aspects (pragmatics, humor, dialect) were excluded.

The distinction between Tone and Localized Tone sometimes appears ambiguous (e.g., fairness depends on cultural context). Can you clarify the conceptual boundary between these two dimensions?

How were reward function coefficients selected (e.g., partial reward +0.5, preference bonus +1, format penalty -0.2)? Did you perform sensitivity analysis?

< Suggestions for Improvement >
Strengthen Statistical Rigor: Perform and report statistical significance tests for all major claims. Particularly provide p-values, confidence intervals, and effect sizes for SFT vs. RL, pairwise vs. pointwise evaluation, and cross-language comparisons. Use bootstrapping or permutation tests to obtain robust estimates.

Clarify Causal Mechanisms: Conduct additional experiments revealing why pairwise evaluation is superior. Include presenting same response twice, systematically varying response order, and analyzing correlation between response quality differences and performance improvements.

Improve Baselines: Add SFT baseline with reasoning to fairly compare with RL. Also include other RL algorithms like DPO or PPO to demonstrate GRPO is best.

Compare with Existing Metrics: Compare MENLO with established multilingual metrics like BLEURT, COMET, BERTScore. Include correlation, relative strengths/weaknesses, and language-specific performance analysis.

---

> ### Author Response · Authors · 2025-11-20
> **Re: controls, statistical significance, quality metric comparison, RL comparison,10 language human eval, SFT + reasoning**
>
> > control experiments
>
> We conducted a control experiment presenting the same response twice using the pairwise template.
>
> - Consistency: Models assigned identical scores >94% of the time
> - Performance: Grading quality dropped significantly, e.g., Qwen3-4B Accuracy fell from 57.1% (A vs. B) to 34.6% (A vs. A)
>
> This confirms our hypothesis (S. 3.1,l. 247-250): the pairwise benefit stems from comparative anchoring against a distinct candidate, not merely the template format
>
> | Models | Consistency (Identical Scores) | Macro_F1 | Preference_acc |
> |---|---|---|---|
> | Qwen3-4B | 99.52 | 25.73 | 34.63 |
> | Qwen3-32B | 99.66 | 31.60 | 36.13 |
> | Llama4-Scout | 94.50 | 28.66 | 43.45 |
> | o3 | 98.36 | 31.43 | 44.70 |
> | gpt-4.1 | 99.97 | 30.81 | 42.70 |
>
> > Are observed differences statistically significant?
>
> Detailed gains and CIs are in Appendix F based on boostramping resampling (5k iterations), demonstrating that improvements for our three primary claims are statistically robust:
>
> 1. RL vs. SFT: RL yields significant gains over the SFT baseline in both Preference Accuracy and Macro-F1 (p≈0).
> 2. Pairwise vs. Pointwise: Pairwise evaluation significantly outperforms Pointwise across all model families (p≪0.001).
> 3. Impact of Rubrics: Grading rubrics provide a consistent boost to grading quality (Macro-F1) across models.
>
> Multiple Testing: p-values for these primary claims are infinitesimal (p≈0, zero negative outcomes in 5k iterations; see Appendix G). Results are highly significant even under strict corrections (e.g., Bonferroni).
>
> > no comparison with other quality metrics
>
> BLEURT and BERTScore are reference-based metrics (not suitable in this setting); COMET and MetricX are specialized for translation.
>
> Spearman correlation between reference-less MetricX24 (lower is better) and our human annotations is -0.07 (p-value 1.88), which indicates that translation-optimized metrics are not directly suitable for general response quality evaluation.
>
> We hope our annotated data can enable the development of new multilingual quality metrics beyond translation.
>
> > no comparison with other RL method or rewards
>
> We investigate judge settings (pointwise, pairwise, zero-shot, few-shot, with/without rubrics, SFT/RL-trained). For SFT and RL, we focus on two representative settings. We also explicitly ablated different reward designs (binary, smoothed, preference, combined) in S. 4.3, Table 6. More extensive details of RL design are not our focus.
>
> > Human evaluation only on 10 languages
>
> We collected 81,014 human annotations in 47 languages. For efficiency, we restrict the validation of the post-trained model to a subset of 10 widely used languages. We think this is a reasonable trade-off, given the large total number of annotations.
>
> > SFT with reasoning included
>
> We agree an SFT baseline with human reasoning would be ideal. However, MENLO, like many real-world preference datasets, does not contain these human reasoning traces; it only provides human-annotated 1-5 scores.
>
> This data limitation is the main motivation for our RL-based approach. The SFT baseline represents the standard method for this 'labels-only' data. Our RL framework is specifically designed to overcome this exact problem: it learns to generate reasoning and assign accurate scores without needing human reasoning traces as supervision (as mentioned in lines 328-329).
>
> Our experiment is designed to show that RL is a superior method for this common data scenario, not to create an "unfair" comparison.
>
> > justification for model selection
>
> We perform zero-shot experiments with a wide range of models. For fine-tuning, we choose two representative, practically small open-weight models that cover different model families, architectures (dense vs MoE), and sizes.
>
> > rationale for four dimensions
>
> The dimensions correspond to key characteristics of a native-level response:
> - fluent: corresponds to linguistic acceptability and language proficiency;
> - factual
> - aligned with the target culture (Localized Tone): includes humor, idiomatic dialectal language, etc
> - generally useful as an assistant (Tone): includes pragmatics such as implicit intent understanding
>
> Some Tone aspects can exhibit locale-specific variation, which annotators take into consideration.
>
> > reward function coefficient selection
>
> - Format (-0.2): A positive reward (+1.0) degraded performance as the model prioritized this "easy" task over complex reasoning. We thus switched to a small negative penalty
> - Smoothing (+0.5): A partial reward for "near misses" provides a dense signal to stabilize training
> - Preference (+1.0): Preference accuracy is treated as a distinct multitask objective; 1.0 is selected to represent a straightforward, unweighted summation
>
> Full grid search was computationally infeasible; Table 6 serves demonstrates that our combined reward design outperforms simpler variations (e.g., +2.3% Macro-F1 over Binary Only)
>
> ---
>
> If we’ve addressed your concerns, would you consider increasing your score?

---

### Official Review · Reviewer_M539 · 2025-11-01

**Soundness:** 4
**Presentation:** 2
**Contribution:** 4
**Rating:** 8
**Confidence:** 5

**Summary:**

This works proposes a dataset and a LLM-judge based evaluation framework to evaluate whether an LMs' response is native-like quality. Specifically, the authors attempt to assess four aspects: (1) language quality/coherence, (2) alignment with cultural/lignuistic nuance, (3) factual correctness, (4) overall writing style and helpfulness. The dataset (Menlo) spans across 47 languages and includes 6423 instances. Then, the authors evaluate different models in point wise and pair wise settings when functioning as a judge to evaluate responses. Also, the authors train a judge model with RL and show that it could match or outperform the performance of Qwen3-32B and GPT-4.1. The authors also show that this judge model could also work as a generative reward model to improve LMs' ability to write fluent responses.

**Strengths:**

1. The Menlo dataset was annotated with 1934 annotators and also has high krippendorff's alpha value, indicating that it is a very high-quality dataset.

2. The list of experiments is very extensive: starting from gathering a dataset, training a judge, and then using the judge to train a generator. Every step is supported by a experimental result, which is very helpful for future work in this area.

**Weaknesses:**

1. Considering that the prompts consisting the Menlo dataset were derived using a prompt template, it could be possible that the prompts are synthetic in nature. There should be an explanation to which extent these prompts mirror what non-English speakers might prompt to frontier models in real-world scenarios. One possible way would be to gather the prompts on WildChat (that are non-English) and measure how similar the two groups are compared to other multilingual meta-eval benchmarks such as M-RewardBench or MM-Eval.

2. The fact that "pairwise ranking" yields higher performance compared to "pointwise" is somewhat trivial - the caveat should be that it incurs higher inference costs. Yet, the paper simply concludes that pairwise has higher performance compared to pointwise. For a fair comparison, since pointwise is O(N) and pairwise is O(N^2), it is possible to prompt the LLM judge N times and apply either score averaging [1,2] or self-consistency [3].

3. In Table 4, it shows that adding Rubrics help the models to predict better judgments. When looking into the rubrics in Appendix A.3., these are explanations of each criteria, not instance-specific rubrics [3]. I think the term "context-specific rubrics" should be adjusted to "detailed rubrics" for readers to not confuse with instance-specific rubrics. Also, it would be nice to add a qualitative example of how the chain-of-thoughts generated by the judges differ when including or excluding the rubrics.

4. In Figure3, there are some languages that benefit more and other languages that do not. What is the difference? Is it the limitation of RL training (that means there should be more pretraining for some languages) or is there another reason behind it that the authors conjecture?

5. Looking at the one of the qualitative example between Figure 9-16 (I won't specify for anonymity), I'm not sure if the grades make sense, which makes me a bit skeptical if the large group of annotators were instructed and managed properly. Can you give me details of how these people were recruited, how much they were incentivized, and how they were qualified?

[1] Kim, S., Shin, J., Cho, Y., Jang, J., Longpre, S., Lee, H., ... & Seo, M. (2023, October). Prometheus: Inducing fine-grained evaluation capability in language models. In The Twelfth International Conference on Learning Representations.
[2] Kim, S., Suk, J., Longpre, S., Lin, B. Y., Shin, J., Welleck, S., ... & Seo, M. (2024, November). Prometheus 2: An Open Source Language Model Specialized in Evaluating Other Language Models. In Proceedings of the 2024 Conference on Empirical Methods in Natural Language Processing (pp. 4334-4353).
[3] Kim, S., Suk, J., Cho, J. Y., Longpre, S., Kim, C., Yoon, D., ... & Seo, M. (2025, April). The BiGGen Bench: A Principled Benchmark for Fine-grained Evaluation of Language Models with Language Models. In Proceedings of the 2025 Conference of the Nations of the Americas Chapter of the Association for Computational Linguistics: Human Language Technologies (Volume 1: Long Papers) (pp. 5877-5919).

**Questions:**

1. In Table 2, Menlo is displayed with some benchmarks (e.g., MM-Eval and M-RewardBench), it might be good to specify the number of instances used as benchmarking (1776 instances).

2. The Biggen-bench [1] also includes a multilingual category with 420 human judgments, it would be great if the authors could add it to Table 2.

[1] Kim, S., Suk, J., Cho, J. Y., Longpre, S., Kim, C., Yoon, D., ... & Seo, M. (2025, April). The BiGGen Bench: A Principled Benchmark for Fine-grained Evaluation of Language Models with Language Models. In Proceedings of the 2025 Conference of the Nations of the Americas Chapter of the Association for Computational Linguistics: Human Language Technologies (Volume 1: Long Papers) (pp. 5877-5919).

---

> ### Author Response · Authors · 2025-11-20
> **Re: prompt comparison, pairwise ranking complexity, CoT judge example, discussion of RL gains, annotator overview**
>
> Thank you for the kind and insightful comments.
>
> > There should be an explanation to which extent these prompts mirror what non-English speakers might prompt to frontier models in real-world scenarios.
>
> Below, we compare WildChat, MENLO, MM-Eval, and M-RewardBench prompts regarding their response length (in characters). We focus on the five languages that are present in all datasets: Spanish, French, Chinese, German, and Russian. For MM-Eval and M-RewardBench, we select the Chat subsets as other subsets (such as reasoning, etc) are not representative of chat conversations. We calculate the median prompt length per language as some datasets such as WildChat contain extreme outliers, which affect the mean.
>
> |  | es | fr | zh | de | ru |
> |---|---|---|---|---|---|
> | wildchat | 108 | 86 | 24 | 10311 | 108 |
> | menlo | 161.5 | 170 | 42 | 174.5 | 148 |
> | mm-eval | 76.5 | 77 | 20 | 101 | 54 |
> | m-rewardbench | 133 | 151.5 | 35 | 143.5 | 129 |
>
> MENLO’s prompt length is slightly higher than that of other datasets as prompts integrate additional locale-specific information in the request.
>
> While all datasets include chat-related prompts, prompts in MENLO focus on locale-specific and cultural information.
>
> > The fact that "pairwise ranking" yields higher performance compared to "pointwise" is somewhat trivial - the caveat should be that it incurs higher inference costs. [...] pointwise is O(N) and pairwise is O(N^2)
>
> We would like to disagree. I think the reviewer might have misunderstood our pairwise ranking evaluation. While the pointwise judge needs to be applied to every response out of N responses, the pairwise judge assesses pairs of responses, i.e., it is applied N/2 times. So the pointwise judge has O(N) complexity and the pairwise judge has O(N/2) complexity (not O(N^2)).
>
> > I think the term "context-specific rubrics" should be adjusted to "detailed rubrics" for readers to not confuse with instance-specific rubrics
>
> Thanks for the suggestion. We have updated the wording in the revised version of the paper.
>
> > Also, it would be nice to add a qualitative example of how the chain-of-thoughts generated by the judges differ when including or excluding the rubrics.
>
> We thank the reviewer for this suggestion. When including rubrics the judge generation is on average 15-20% longer with added content of referencing the rubrics definition before the final grading. We have added examples in Appendix F (page 40) to reflect Qwen3 models and API models generation difference in a pairwise setup. We will expand this section in the final version to include more examples.
>
> > In Figure3, there are some languages that benefit more and other languages that do not. What is the difference? Is it the limitation of RL training (that means there should be more pretraining for some languages) or is there another reason behind it that the authors conjecture?
>
> We acknowledge that analysing the language-wise performance benefit from training provides more insights. In Table 23 (Appendix), we provided the Zero-shot (ZS) and RL-trained preference accuracy per language. To diagnose the impact of pre-training limitations versus challenges in the RL optimization, we distinguish the following four categories:
>
> | Category | Example Languages | Conjectured Primary Limitation |
> |---|---|---|
> | (1) Low Zero-Shot, High RL | gu_IN (+16.21), pt_PT (+10.52) | The base model lacked alignment in the downstream domain, but has sufficient linguistic foundation, and the high-quality RL data successfully improved. |
> | (2) Low Zero-Shot, Low RL | bn_BN (-3.45), da_DK (+7.69) | Pre-training Coverage. While the RL phase may provide gains (e.g., +7.69 for da_DK), the overall absolute score remains low, confirming that the limited language coverage in the pre-training data is the limiting factor. |
> | (3) High Zero-Shot, Low RL | ar (-2.63), fr_FR (-2.64) | Optimization Trade-offs. Challenge in multitask/multilingual training that some languages may regress in performance (the "win some, lose some" effect). |
> | (4) High Zero-Shot, High RL | tr_TR (+10.26), ko_KR (+13.16) | Optimal performance from strong linguistic foundation and effective RL alignment. |
>
>  > Can you give me details of how these people were recruited, how much they were incentivized, and how they were qualified?
>
> We have added an overview of annotator training, etc to Appendix B (page 30).
>
> > The Biggen-bench [1] also includes a multilingual category with 420 human judgments, it would be great if the authors could add it to Table 2.
> > In Table 2, Menlo is displayed with some benchmarks (e.g., MM-Eval and M-RewardBench), it might be good to specify the number of instances used as benchmarking (1776 instances).
>
> Thanks for the suggestion. We’ve added Biggen-bench and the size of MENLO’s test split to Table 2. Let us know if we’ve missed anything else.

---

### Official Review · Reviewer_yzH6 · 2025-11-01

**Soundness:** 3
**Presentation:** 3
**Contribution:** 3
**Rating:** 6
**Confidence:** 3

**Summary:**

This manuscript contributes to LLM-as-Judge by proposing MENLO, building a 6k+ human-annotated dataset, establishing four quality evaluation dimensions, showing RL's impact on LLM-as-judge and the dataset's benefit to multilingual RL, though the RL-enhanced LLM-as-judge still lags behind human annotation.

**Strengths:**

- The paper plans to open-source its dataset. A high-quality training dataset is currently in urgent need by the open-source community.
- The paper’s dataset primarily contributes to the improvement of LLM-as-Judge — a critical need for the current RL paradigm, and a highly important topic that requires further advancement.

**Weaknesses:**

The multilingual section of the paper is derived from direct translation, which may lack native-specific traits. For instance, the level of attention paid to art can vary across different countries, and this discrepancy may introduce a certain degree of bias.

**Questions:**

none

---

> ### Author Response · Authors · 2025-11-20
> **Re: Translation-based prompts lacking native-specific traits**
>
> Thank you for the kind review.
>
> Regarding your concern:
>
> > The multilingual section of the paper is derived from direct translation, which may lack native-specific traits.
>
> Our prompts derive from English templates to ensure coverage consistency across a large number of languages. While localization includes cultural adaptation, we acknowledge that independently authored prompts in each language would better capture native discourse structures. We’ve added a note highlighting this trade-off to the revised version of the paper (lines 711-714).
>
> We note that we focus most of our efforts on ensuring annotation quality of the responses, which are generated directly in the target language and assessed by native speakers based on their native-specific traits.
>
> ---
>
> If we’ve addressed your concern, given that this is the only weakness mentioned in your review, would you consider increasing your score?

---

> > ### Comment · Reviewer_yzH6 · 2025-11-26
> >
> > Your response has partially addressed my concerns.
> > I believe the paper is sufficiently solid to merit acceptance, but its novelty and inspiration may be limited.
> >
> > Given that this submission already has a very high average score, I have decided to uphold this rating.

---

### Official Review · Reviewer_kRGr · 2025-11-03

**Soundness:** 3
**Presentation:** 3
**Contribution:** 3
**Rating:** 6
**Confidence:** 3

**Summary:**

This paper introduces MENLO, a framework for evaluating the native-like quality of multilingual LLM responses based on audience-design principles. MENLO defines four core dimensions, Fluency, Tone, Localized Tone, and Localized Factuality and provides localized prompt templates to elicit culturally grounded responses.
MENLO further contributes a large scale multilingual human evaluation dataset comprising 6,423 prompt-response pairs with 81,014 annotations spanning 47 languages/varieties, achieving strong inter-annotator agreement (avg. Krippendorff’s α = 0.84).
The paper empirically demonstrates the superiority of pairwise over pointwise judgments for aligning evaluator preferences, and shows that incorporating structured rubrics improves pointwise reliability.
Additionally, the authors explore LLM-Judge training methods, including multi-task reinforcement learning and reward shaping to boost automatic judge models closer to human-level performance.
Finally, the trained judges are shown to be effective as generative reward models, improving policy models’ multilingual proficiency. However, the study highlights that LLM-based evaluations can overestimate model performance relative to human raters, revealing an important limitation.

**Strengths:**

- Extensive Multilingual Coverage Demonstrates strong value by tackling the critical aspect of native-like linguistic quality, incorporating human validation across a broad set of languages.
Provides substantial experimental diversity with 47 languages/varieties, 6,423 prompt–response pairs, and 81,014 annotations, ensuring wide multilingual representation.
- Well-Structured and Systematic EvaluationOffers quantitative evidence that zero-shot pairwise evaluation consistently outperforms pointwise evaluation in both Macro-F1 and preference accuracy.
Empirically validates the practical utility of rubric-guided evaluation showing benefits for both absolute and comparative assessments, indicating strong applicability in real-world evaluation pipelines.

**Weaknesses:**

- MENLO uses localized prompts and employs translators/annotators from the corresponding regions, there still exist diverse linguistic variations within a single locality across social groups, age ranges, and socioeconomic classes. Therefore, constructing gold labels based on a single demographic standard may introduce bias toward a particular perspective.
- From a user perspective, even if a response appears native-like, its practical utility drops significantly if it is factually incorrect especially with respect to localized factual knowledge. If the reward model does not account for factuality, the system may optimize for style while increasing the risk of conveying incorrect information.

**Questions:**

1. Do the four evaluation dimensions (Fluency, Tone, Localized Tone, Localized Factuality) include moral or behavioral conventions that are specific to certain regions or cultures?
2. Pairwise evaluation involves comparing two responses at a time. Do the authors plan to conduct multi-sample comparison experiments, examining whether the presence or absence of rubrics affects performance?

---

> ### Author Response · Authors · 2025-11-20
> **Re: "Nativeness" beyond demographics, factuality reward modeling, culture-specific conventions, rubric experiments**
>
> Thank you for the kind review and insightful feedback.
>
> > constructing gold labels based on a single demographic standard may introduce bias toward a particular perspective.
>
> We recognize the challenge of language diversity and fluidity: any attempt to exhaustively define a language’s “nativeness” will fall short as contexts change. We take a deliberate set of measures to ensure that nativeness is evaluated and examined within interactional contexts that are more nuanced than demographic categories:
>
> **Prompt design.** When designing human-written prompts, we focus on expandability and customizability so they can accommodate different dialects, subcultures, and interactional contexts. The prompts are also often embedded in social situations rather than isolated sentences. For example:
>
>  “Imagine you’re at an [locale_nationality] friend’s family gathering in [locale_country]. How would you politely ask for a second helping of food?”
>
> **Recruitment.** We prioritize contributors with multilingual backgrounds and familiarity with diverse language styles—across dialects, age groups, and subcultural varieties.
>
> **Rubric design.** For rubrics involving local knowledge or localized tone, we reward responses that reflect diverse perspectives and penalize overly reductive generalizations or stereotypes.
>
> **Annotator training.** Annotators are encouraged to rate responses based on how well they fit the given interactional context, rather than evaluating them against a presumed demographic norm.
>
> **Reliability.** Each response is annotated by at least three annotators to reduce individual subjective bias and ensure more robust, context-informed ratings.
>
> We acknowledge that evaluating nativeness remains inherently subjective and inevitably shaped by individual perspectives. This is why the MENLO framework is designed with templative flexibility and extensibility: it can incorporate community input, accommodate additional contextual variations, and evolve as language practices evolve. We believe this data is useful to the community towards the study of multilingual pluralistic alignment.
>
> > If the reward model does not account for factuality, the system may optimize for style while increasing the risk of conveying incorrect information
>
> We agree that assessing factuality of long-form responses in a massively multilingual setting remains a challenge. We acknowledge this in lines 345-350, 1647–1649. We note that many prompts in the Tone and Localized Tone categories contain writing, advice, and idiomatic queries where there is less room for factual errors.
>
> For reliable open-book multilingual factuality evaluation, we would need to break down responses into atomic claims, retrieve relevant web documents, and verify claims, which is beyond the scope of this work. As far as we are aware, this pipeline has only been applied to the narrow biography domain using Wikipedia (Multi-FAct).
>
> To our knowledge, MENLO is one of the only multilingual factuality datasets with human-annotated long-form responses and so can enable new research in this area.
>
> > Do the four evaluation dimensions (Fluency, Tone, Localized Tone, Localized Factuality) include moral or behavioral conventions that are specific to certain regions or cultures?
>
> The annotation instructions include assessment of cultural relevance (how well does the model’s tone align with local values, customs, and norms?) for Localized Tone and relevance (is the response relevant to the local context, and written from a local point of view?) for Localized Factuality. We aimed to avoid prescribing region or culture-specific moral or behavioral conventions, asking the annotators to use the norms and conventions of their cultural and local background as part of their assessment instead.
>
> > Do the authors plan to conduct multi-sample comparison experiments, examining whether the presence or absence of rubrics affects performance?
>
> Thank you for this interesting suggestion. In Section 3.2 (lines  262-265), we analyzed the impact of rubrics and found a clear trend: rubrics provided a substantial boost in pointwise evaluation (+4.3% Macro-F1) but only marginal gains in pairwise evaluation (~1%).
>
> As noted in lines 266-268, we hypothesize that comparing multiple responses provides a strong intrinsic "grounding" signal, which reduces the model's reliance on explicit grading criteria.
>
> This experiment can be theoretically extended to multi-sample comparisons where k>2, however current annotated data only consists of pairwise labels. We hypothesize that the trend would continue: as k increases, the marginal benefit of rubrics would likely decrease further or plateau. We agree this is a valuable direction for future work.
>
> ---
> Thanks again for the helpful questions and feedback. If we’ve addressed your concerns, would you consider increasing your score?

---

### Meta-Review · Area_Chair_cDz9 · 2026-01-07

**Summary:**

The paper presents **MENLO**, a comprehensive framework and dataset (6,423 human-annotated pairs across 47 languages) designed to evaluate and enhance the "native-like" quality of LLM responses. The work is grounded in sociolinguistic theory, specifically "audience design," and decomposes quality into four dimensions: Fluency, Tone, Localized Tone, and Localized Factuality.



The reviewers generally commended the **scale and quality of the dataset** (Krippendorff's ) and the **extensive experimental pipeline**, which transitions from evaluation to active model improvement via RL-trained judges acting as reward models. While initial concerns were raised regarding statistical significance, the inherent biases of defining "nativeness," and the computational complexity of pairwise comparisons, the authors provided a robust rebuttal. The collective feedback suggests the paper provides a valuable, high-quality resource for the multilingual alignment community.

**Reviewer Concerns:**

**Concerns Addressed by the Rebuttal:**
- **Statistical Rigor:** The authors provided p-values and confidence intervals using 5,000 bootstrap iterations, confirming that the gains from RL and the superiority of pairwise evaluation are statistically robust.

- **Pairwise Complexity:** The authors clarified that their pairwise setup is  rather than , making it computationally efficient and comparable to pointwise methods.

- **Causal Mechanisms:** A control experiment (A vs. A comparison) was conducted, demonstrating that the benefits of pairwise evaluation stem from comparative anchoring rather than the prompt template alone.

- **Terminology and Qualitative Data:** The authors updated the term "context-specific rubrics" to "detailed rubrics" and added Chain-of-Thought examples to the appendix to illustrate how the model utilizes these rubrics.


**Outstanding or Partially Addressed Concerns:**

- **Localized Factuality:** While the authors acknowledge that assessing factuality is difficult in a massively multilingual setting, the risk remains that models might optimize for linguistic style at the expense of factual accuracy.

- **Prompt Naturalness:** There is an ongoing concern that using English templates for prompt generation (even with localization) might miss native-specific discourse structures.

- **Subjectivity of Nativeness:** Defining "nativeness" against a single demographic standard can introduce bias. Although the authors used multiple annotators and flexible prompts, the inherent subjectivity of the task remains a fundamental limitation.

**Reviewer Scores:**

Based on the discussion quality and the authors' detailed responses, I estimate the following score changes if a full discussion had been finalized:

**Reviewer KRGr (Initial: 6):** **Remains 6.** The reviewer’s concerns regarding factuality and demographic bias are conceptual and acknowledged by the authors as scope limitations.

**Reviewer yzH6 (Initial: 6):** **Remains 6.** During the discussion, this reviewer explicitly stated they would "uphold this rating," noting that while concerns were partially addressed, they perceived the novelty to be limited.

 **Reviewer M539 (Initial: 8):** **Remains 8.** The reviewer was already highly positive. The authors' clarification on  complexity and the addition of CoT examples likely solidified this high score.

 **Reviewer kNrK (Initial: 6):** **Likely 7 or 8**. This reviewer had the most technical critiques regarding statistical rigor and causal analysis. Given the authors' exhaustive response with new control experiments and infinitesimal p-values, a significant score increase is highly probable.

---

### Decision · Program_Chairs · 2026-01-26

Accept (Poster)